



**A case study of the large-scale traveling ionospheric disturbances in the East Asian**
**sector during the 2015 St. Patrick's Day geomagnetic storm**
Jing Liu[1], Dong-He Zhang[1]*, Anthea J. Coster[2], Shun-Rong Zhang[2], Guan-Yi Ma[3], Yong-Qiang
Hao[1], Zuo Xiao[1].
1, Department of Geophysics, Peking University, Beijing, China, 100871
2, MIT Haystack Observatory, Westford, Massachusetts, USA
3, National Astronomical Observatories, Chinese Academy of Sciences, Beijing, China
**Abstract**
This study gives the first observation of the large-scale traveling ionospheric disturbances
(LSTIDs) in the East Asian sector during the 2015 St. Patrick's Day (March 17, 2015) geomagnetic
storm. For the first time, 3 dense networks of GPS receivers in China and Japan are combined
together to obtain the 2-dimensional (2D) vertical total electron content (VTEC) perturbation maps
in a wider longitudinal range than previous works in this region. Results show that a negative LSTID
spanning at least 60° in longitude (80°E-140°E) occurs and propagating from high to lower latitudes
around 09:40-11:20 UT. It is followed by a positive LSTID which shows a tendency of dissipation
starting from the East side. The manifestation of the 2D VTEC perturbation maps is in good
agreement with the recordings from 2 high-frequency Doppler shift stations and the iso-frequency
lines from 8 ionosondes. Then, the propagation parameters of the LSTIDs are estimated by applying
least square fitting methods to the distinct structures in the 2D VTEC perturbation plots. In general,
the propagation parameters are observably longitudinal dependent. For example, the propagation
direction is almost due southward between 105°E-115°E, while it is slightly South by West/East in
the West/East side of this region. This feature is probably related to the regional geomagnetic
declination. The mean values of the period, trough velocity ($V_t$), crest velocity ($V_c$), and wavelength
of the wavelike LSTIDs in the studied longitudinal bands are $74.8 \pm 1.4$ minutes, $578 \pm 16$ m/s, $617$
$\pm 23$ m/s, and $2691 \pm 80$ km, respectively. Finally, using the VTEC map data from the Madrigal
database of the MIT Haystack Observatory, the characteristics of the ionospheric disturbances over
the European sector (30°N-70°N, 10°E-20°E) are also studied. The results are very different from
those in the East Asian sector in parameters like the occurrence time, oscillation period, and
propagation velocities.
**Keywords: Geomagnetic Storm; LSTID; GPS TEC.**





**1.  Introduction**

2        During the geomagnetic storm, the solar wind energy is impulsively or continually injected into

the earth polar region and making the atmospheric and ionospheric states deviate greatly from their
background levels [Fuller-Rowell et al., 1994]. In general, the response of the ionosphere to the
geomagnetic storm is classified by a variety of different features, one of which is the large scale
traveling ionospheric disturbance (LSTID). It is the wave-like perturbation mainly propagating
equatorward from high latitudes that is considered to be the manifestation of the presence of
atmospheric gravity waves (AGWs) within the ionosphere caused by Joule heating or Lorenz-drag
forcing in the Auroral regions during geomagnetic storm period [Hines, 1960; Richmond and Roble,
1979; Hocke and Schlegel, 1996].

11       In earlier years, the acquisition of the continuous evolution of LSTIDs on a global scale was

limited by the availability of the ionospheric observations. In order to obtain the propagation
characteristics of LSTIDs on a large spatial scale, researchers needed to organize their findings from
limited ionospheric observations, for example, the foF2 data from sparsely distributed ionosondes.
In the 1980s, the GPS method was introduced into the ionospheric study [Klobuchar, 1986; Lanyi
and Roth, 1988; Coster and Gaposchkin, 1989]. With the dense and worldwide distributed GPS
receivers, some characteristic ionospheric phenomena, like traveling ionospheric disturbances
(TIDs) [Saito et al., 1998; Tsugawa et al., 2004; Ding et al., 2007], ionospheric storms [Ho et al.,
1996], and ionospheric responses to solar flares [Afraimovich, 2000a; Zhang and Xiao, 2005] were
revisited frequently and new results were obtained.

21       The propagation characteristics of LSTIDs are always topics of great research interest

[Hunsuncer, 1982; Ho et al., 1996; Balthazor and Moffett, 1999; Afraimovich et al., 1998, 2000;
Shiokawa et al., 2002; Tsugawa et al., 2003, 2004; Ding et al., 2008, 2014; Jakowski et al., 2008;
Borries et al., 2009, 2017; Habarulema et al., 2015, 2016; Zakharenkova et al, 2016; Figueiredo et
al., 2017; Pederick et al., 2017; Cherniak et al., 2018; Lyons et al., 2019]. Based on limited GPS
stations measurements, Afraimovich et al. [1998] propose a radio interferometry method to roughly
estimate horizontal propagation velocities and phase front angles of TIDs. Further, the world-wide
or local dense distribution of the GPS receivers networks facilitates the acquisition of the global or
regional TEC perturbation maps with high spatial and temporal resolutions to reveal the detailed
propagating characteristics of TIDs [Ho et al., 1996; Saito et al., 1998; Tsugawa et al., 2004; Borries
et al., 2009; Ding et al., 2012]. With more than 60 GPS receivers distributed worldwide, Ho et al.
[1996] studied the global distribution of TEC variations and perturbations during a magnetically
disturbed period. They identified a TID propagating from the northern sub-auroral region to lower
latitudes at a speed of about 460 m/s. The GPS Earth Observation Network (GEONET) in Japan is
one of the densest GPS receiver networks on the Earth, and utilizing its data two-dimension (2D)
TEC perturbations over Japan can be mapped. With these high-resolution TEC perturbation maps,
the spatial structures and temporal evolutions of a TID in the nighttime mid-latitude ionosphere over
Japan were revealed clearly [Saito et al., 1998]. Since then, with this dense GPS network, the
characteristics of LSTIDs over Japan are carefully studied through case and statistical analysis, and
some propagation features of TIDs in this region are revealed [Saito et al., 2001; Shiokawa et al.,
2002; Tsugawa et al., 2003, 2004, 2006].

42       For the LSTID with scales of thousands of kilometers, the extensive spatial coverage of

ionospheric observations is undoubtedly useful for capturing its propagation features. In recent years,
the GPS data from densely distributed GPS stations in China are used to study LSTIDs in this region





[Ding et al., 2012, 2013, 2014; Song et al., 2013]. Based on the GPS data from the Crustal Movement Observation Network of China (CMONOC), Ding et al. [2012] obtains temporal continuous 2D imaging of ionospheric disturbances during the geomagnetic storm on May 28, 2011, and find two LSTIDs moving southwestward with the front width of at least 1600 km during different storm phases. In addition, through the comparative climatological study of LSTID over North America and China, the different time dependence of LSTID occurrence over two longitudinal sectors were revealed statistically [Ding et al., 2014]. These studies further emphasize the effectiveness of the large coverage, high-resolution ionospheric observations from GPS networks on the detailed investigation of the ionospheric disturbances structures.

The propagating direction of the LSTID during the geomagnetic storm has always been focused on for the LSTID studies. From case and statistical studies about LSTID during geomagnetic storm period over East-Asia region conducted independently by Chinese and Japanese scientists in recent years, the dominant propagating direction of LSTID in China and Japan is a little different. It mainly propagates South by West in China region [Ding et al., 2014], while it mainly propagates South by East in Japan region [Tsugawa et al., 2004]. Although the geomagnetic declination is considered to be one of the main factors to be responsible for the propagating direction of LSTID based on different LSTID studies, the LSTID studies concerning the same geomagnetic storm using both China and Japan GPS networks together have not yet been reported.

During the period of 17–18, March 2015, a large geomagnetic storm occurred, which is the strongest one in the 24th solar cycle. During this storm, LSTIDs over European and American sectors are detected and analyzed with data from Global Navigation Satellite Systems (GNSS) stations [Zakharenkova et al., 2016]. Meanwhile, two high frequency (HF) Doppler stations operated by China Meridional Project [Wang, 2010] at mid-latitude China record a large ionospheric HF Doppler shift after 10:00 UT, which appears to indicate LSTIDs related to the geomagnetic storm. In this study, the multi-network of densely distributed GPS receivers and an ionosonde network will be used to reveal the propagating characteristics of these large disturbances recorded in HF Doppler shift receivers in the East Asian region, especially the characteristics of the dominant propagating direction of this disturbance over China and Japan.

## 2. Data and Methods

Figure 1 illustrates the locations of ground-based receivers that are used in this study from 4 GNSS networks distinguished by colors. They are Chinese Meteorological GNSS Network (CMGN), CMONOC in China, GEONET in Japan, and International GNSS Service (IGS). These receivers are selected through data quality checking and regional restriction (10°N ~ 60°N, 70°E ~ 150°E), and the numbers of used stations are 259, 220, 1300, and 31 for CMGN, CMONOC, GEONET, and IGS, respectively. The sample rate of all GPS data is 30 seconds. Combining the carrier phase and pseudo-range measurements in two L-band frequencies of GPS receivers' observations, the vertical TEC can be obtained. In the calculation, the height of the ionospheric thin shell is set to be 400 km, and the cutoff elevation angle is 30 degrees. The detailed process of the TEC calculation from GPS data can be found in our previous works [Zhang et al., 2009; Zhang et al., 2010].

Different methods have been used for extracting the TEC perturbations related to LSTIDs in previous works [Wan et al., 1997; Afraimovich et al., 2000; Shiokawa et al., 2002; Nicolls et al., 2004; Tsugawa et al., 2004; Ding et al., 2007]. Afraimovich et al. [2000] suggest that the LSTID

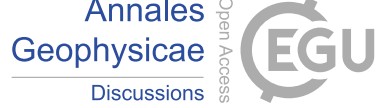



characteristics in TEC can be determined by removing the trend with 3 to 5 order polynomials in
order to eliminate the trends introduced by the motion of satellites and variations of the regular
ionosphere. For a similar purpose, Shiokawa et al. [2002] subtract a running average of TEC over 1
hour from the raw TEC. With more than 1000 GPS receivers over Japan, a series of 2D TEC
perturbation maps can be obtained. Ding et al. [2007] develop another method of obtaining the 2D
TEC perturbation maps by expressing the vertical TEC as a one-order function of local time and
latitude. According to their argument, this method is sufficient to remove background trends for
continuous observation of a GPS receiver-satellite pair without introducing artificial perturbations.
After comparing the results of these methods, a method similar to Ding et al. [2007] is conducted in
this study, in which the vertical TEC at an ionospheric pierce point (IPP) is treated as a function of
universal time (UT), longitude (Lon), and latitude (Lat), i.e.,
$$VTEC_0 = C_0 + C_1 UT + C_2 Lon + C_3 Lat$$
$$VTECP = VTEC - VTEC_0$$
in which $VTEC_0$ is the background change and VTECP is VTEC perturbation. Then, the obtained
VTECP data is reorganized into pixels which are bounded by 10°N ~ 60°N, 70°E ~ 150°E and with
a spatiotemporal resolution of 1° longitude × 1° latitude × 10 minutes. The VTECP value for each
pixel is set to be the average of all VTECP data of which the IPP and UT locate in this pixel. After
these steps, the featured ionospheric disturbances are expected to appear on a series of 2D VTECP
maps.
As a comparison, the VTEC map from Madrigal database of the MIT Haystack Observatory
(http://cedar.openmadrigal.org/) is used to reveal the ionospheric disturbances in the European
sector (30°N ~ 70°N, 10°E ~ 20°E). This database provides worldwide VTEC values in 1° latitude
× 1° longitude pixels with a temporal resolution of 5 minutes [Rideout and Coster, 2006] and has
good data coverage in European and American sectors. VTEC maps with such a high spatiotemporal
resolution are suitable to reveal the structures of traveling ionospheric disturbances [Zhang et al.,
2017].
The Doppler shift data observed at two high frequency (HF) Doppler sounding stations in
China is collected, of which the station codes are MDT (40.4°N, 116.9°E), and SZT (22.6°N,
114.1°E). The sounding system continuously receives electromagnetic waves with a stabilized
frequency of 10 MHz transmitted by the National Time Service Center (NTSC) (35.7°N, 109.6°E)
to detect the ionospheric disturbances through the Doppler shifts of this standard frequency. These
shifts are considered to be caused by ionospheric variations mainly around the reflecting point of
the electromagnetic wave in the ionosphere. According to the geometrical relationships, the
locations of the reflecting point for MDT and SZT are (38.0°N, 113.2°E) and (29.2°N,111.8°E),
respectively. These stations are marked in Figure 1 with colored stars.
In this study, ionograms from 8 ionosonde stations in China middle latitude are used to derive
the iso-frequency lines, which vary as a function of universal time and virtual height. The sample
rate of the ionograms is 15 minutes. These ionosondes belong to the China Research Institute of
Radio-wave Propagation (CRIRP) and their locations are marked in Figure 1 with lime triangles. In
addition, the condition of the geomagnetic storm is shown with data from the high resolution (5
minutes) OMNI dataset, which is downloaded from the FTP service of the NASA Goddard Space
Flight Center (https://spdf.gsfc.nasa.gov).

**3. Results**



### 3.1 Observations

Figure 2 shows the variations of (a) solar wind speed, (b) interplanetary magnetic field (IMF) $B_z$ component, (c) the SYM-H index, and (d) the AE index from the OMNI dataset, and the time range is from 18:00 UT, 16 March 2015 to 06:00 UT, 18 March 2015. It should be noted that the solar wind magnetic field and plasma data are time-shifted to the bow shock nose to better support the solar wind-magnetosphere coupling studies. It can be seen clearly that a geomagnetic storm occurred on 17 March 2015, with the sudden storm commencement (SSC) at ~ 04:45 UT, which is characterized by a sharp increase (marked with vertical dashed lines) in the solar wind speed, Bz, and SYM-H index. The main phase of the storm can be roughly divided into two stages. The first stage is from ~ 06:00 UT, when the IMF Bz component first turns to southward, to ~ 12:00 UT, when the Bz turns southward again after back to northward for about 2 hours. After ~ 12:00 UT, the Bz is southward for the most time, until it enters the recovery phase. The SYM-H and AE indices show a similar two-stage feature as the Bz. The SYM-H decreases after ~ 06:00 UT, reaches the first minimum at ~ 09:30 UT, and increases to a local maximum at ~ 12:00 UT. Then, it gradually decreases with small oscillations and reaches the minimum value of -233 nT at ~ 22:45 UT. Correspondingly, the AE index exhibits the first increase period between 06:00 UT to 12:00 UT, with the maximum intensity of ~ 1000 nT, and the second period between 12:00 UT to 02:00 UT of the next day, during which the AE increases much larger with several peaks. This storm is the strongest one in the 24th solar cycle [Astafyeva et al., 2015].

During the first stage of the main phase, disturbances are observed successively at two HF Doppler receiver stations shown in Figure 1. Figure 3 illustrates the variations of the Doppler shift records at (a) MDT and (b) SZT between 08:00 UT and 14:00 UT on 17 March 2015. It shows that two distinct positive shifts occur at about 10:21 UT and 10:55 UT, respectively. Shortly after, it exhibits two negative shifts but with much smaller amplitudes. Suppose these successive disturbances indicate a propagating perturbation, according to the estimated locations of the reflecting points that mention above and the occurrence time of the two positive peaks, the approximate speed of this perturbation is about 553 m/s. This value is much larger than the speed of the movement of the ionospheric negative storm that usually occurs in the middle latitude due to storm-induced equatorward wind [Buonsanto, 1999], and the ionospheric storm is not serious in the Asian sector during this period [Astafyeva et al., 2015]. Considering the propagating speed and the interval of the positive-negative variations, the recorded perturbations probably reflect an equatorward propagating LSTID in the East Asian sector.

To confirm this, Figure 4 presents a sequence of 2D VTECP maps between 09:40-11:40 UT on 17 March 2015 with the method described above. The grey areas represent the nightside. Note that the raw value of VTECP is converted into VTECP′ with

$$VTECP' = sign(VTECP) * log_{10}(abs(VTECP) + 1)$$

to make it easier to distinguish the regions with positive and negative perturbations. The yellow lines illustrate the least square fitting results for all the negative pixels within certain rectangular areas bounded by longitudes and latitudes. The lime lines are similar but for pixels with values close to zero (see below for example). These two kinds of lines mark the approximate locations of the wavefronts.

A large-scale wavelike perturbation can be seen clearly in Figure 4. The first relatively distinct wave structure emerges during the (d) 10:10-10:20 UT period, while its sign can already be observed as early as (a) 09:40-09:50 UT in the northwest part of China. During (e) 10:20-10:30 UT, a negative



band that across both China and Japan sectors occur between around 30°N-45°N, which gradually
propagates to lower latitudes in the next tens of minutes. During (f) 10:30-1040 UT, the first clear
wavefront of the positive band appears, which also shows an equatorward movement for at least
half an hour. Finally, there seems to be no distinct wave structure following the positive band.
Considering the spatiotemporal characteristics of this perturbation, it can be preliminarily identified
as an LSTID. By the way, it is interesting to note that the positive bands do not extend to the Japan
sector in (h) and (i), and the corresponding VTECP′ amplitudes seem smaller in the East side than
in the West side. This may be probably related to that the Japan sector has already entered the
nightside.
Both the negative and positive bands exhibit more complex variations when they enter the
equatorial ionospheric anomaly (EIA) region between 20°N-30°N. On the one hand, the amplitude
of VTECP′ is relatively larger than those in the higher latitudes. On the other hand, it seems that
the equatorward propagating of the negative band decelerates significantly in this area, which is
especially shown in (g-l). Such complex features are probably related to the various physical
processes in this region. Ding et al. [2012] suggest that LSTIDs experience severe dissipation in
South China region due to viscosity and heat conductivity at low latitudes, which may account for
the weakening of the equatorward propagating of the wavelike structures.
Our observations of the Doppler shift and VTECP′ maps are in good agreement. To show it
clearly, Figure 5 shows the variations of the mean VTECP′ data near the Doppler reflection points
with the same time range of Figure 3. It can be seen that the troughs at around 10:20 UT in (a) and
10:50 UT in (b) correspond well to the two distinct crests in Figure 3. In addition, the variations of
the VTECP′ between 11:00 and 14:00 are also in a good negative correlation with the Doppler shift
observations for each reflecting point. It should be noted that the variation of VTECP′ at the
reflecting point 1 exhibits more variability than that at the reflecting point 0, especially around 09:00
UT, 10:00 UT, and 12:00 UT. Considering that point 1 (29.2°N,111.8°E) is approaching the EIA
region, the causes for VTEC perturbations are more complicated as mentioned above. This feature
is consistent with the observations of the 2D VTECP′ maps in Figure 4.
Ionospheric parameters from ionograms have been commonly used since early TID studies.
Recently, ionograms observed by digital ionosondes with a sampling rate as high as 100 seconds
was used in TID studies [Klausner et al, 2009]. Figure 6 presents the temporal variations of the
virtual height for iso-frequency lines from ionograms. The names and locations of the corresponding
ionosondes are given in each subplot. On the left side, the results of five stations are arranged in
order from high to lower latitudes, and on the right side, it shows the recordings of four stations in
the same latitudinal belt. The corresponding frequencies are also marked for each iso-frequency line.
We can see clearly that a distinct uplift of the virtual height occurs at 09:45 UT at Manzhouli station,
and it gradually moves equatorward from high to lower latitudes (a-e). Meanwhile, the phase
difference is not observed for the stations on the right column. This means that the ionospheric
disturbance roughly moves along the meridian line in this longitudinal sector (around 115°E), which
corresponds to the results of the 2D VTECP′ map. Moreover, although the time resolution of 15
minutes is relatively low, it can still be identified that the crests in the higher iso-frequency lines
appear earlier than those in the lower ones. Such trends (marked with red dashed lines) indicate a
downward vertical phase velocity, which is one of the typical characteristics of TID and AGW [Hine,
1960; Hocke and Schlegel, 1996].



### 3.2 Estimating Propagation Parameters

As preparation for estimating the propagating parameters of this LSTID, Figure 7 shows a detailed example of the wavefront fitting method with the VTECP′ map in Figure 4(g) (10:40-10:50 UT). The reason for choosing this period is that the structure of the wavefront is relatively clear, and the boundary of the positive and negative wave band in the Japan sector can still be partly identified. The lime line is the least square fitting for the lime dots, of which the absolute VTECP′ values are small enough (bottom 5%) among all the dots in a certain region (75°E-140°E, 30°N-40°N). The wave propagating azimuth (marked with arrows) can be estimated with the normal direction of this fitting line. Results are listed in Table 1 in the second column.

It can be seen clearly that the TID moves due South around 110°E, and in the West/East region, the propagation direction is slightly South by West/East. It should be noted that the morphology of this TID is continuously changing as it moves from high to lower latitudes. Although the azimuths are estimated only with the wavefront data during 10:40-10:50 UT, such longitudinal dependence of azimuths corresponds well with other fitting lines in Figure 4(e, f, g, h).

In order to derive the phase speed, period, and wavelength of this LSTID, the time-latitude plots (TLPs) of VTECP′ are obtained for six longitudinal bands, which are marked with dashed rectangles in Figure 7. For each band, the VTECP′ data is averaged along the latitude for every 6 minutes (0.1 hours), and the results as a function of UT and latitude are illustrated in Figure 8. As mentioned before, the variation of VTECP′ in the EIA region is rather complex, so only the values over 30°N (marked with dashed lines) are used to estimate the speed.

As expected, the most distinctive structures in all panels are the pair of negative and positive bands around 10:40 UT, which correspond to the perturbations moving from high to lower latitudes shown in Figure 4. The structures in the 130°E-140°E are not quite clear, which may be due to the lack of data in some parts of this area, but the trough around 10:40 UT can still be identified. To estimate the meridional phase speeds of these perturbation patterns, the linear least square method is used to fit the pairs of troughs and crests. The data points for the linear fitting are marked with white dots, which are the minimum/maximum values along with each latitudinal bin around the negative/positive structures that we focus on. The phase speeds for wave troughs (Vt) and crests (Vc) can be derived based on the slopes of the fitting lines. Moreover, the period of the wave can be estimated through the time interval between the trough and crest in TLPs. In practice, for each longitudinal region, the average of time lags along all latitudinal bins is set to be the half period of the wave in this region. Finally, with the period and speed, the wavelength can be easily determined.

However, those speed, period, and wavelength are the projections on longitudes. After adjusted by the propagation azimuths that calculated above, the final results of the estimated parameters are also listed in Table 1. The longitudinal dependence of these parameters can be seen clearly. The mean values and standard deviations of the period, Vt, Vc, and wavelength are 74.8±1.4 minutes, 578±16 m/s, 617±23 m/s, and 2691±80 km, respectively. It shows that Vt is in good agreement with the result of 553 m/s derived from the Doppler observation. These parameters are typical for an LSTID. Besides, it is interesting to note that the mean Vc is slightly larger than the mean Vt, which seems like the wave behind is pushing that ahead.

As mentioned above, the VTECP′ in the EIA region seems to exhibit different features compared to that in the middle latitude. It can be seen from Figure 8(c) that VTECP′ in this region also shows a periodic variation, but its period seems larger and time duration is longer than the LSTID. These disturbances are probably related to the complex variations of VTEC after 08:00 UT



(around dusk). Besides, the perturbations at 20°N around 12:00 UT and 13:00 UT show patterns of
poleward movement. Ding et al. [2013] have studied the poleward-propagating LSTIDs in southern
China during a medium-scale storm in 2011. They attribute their observations to the excitation of
secondary LSTIDs during the dissipation of primary disturbances from the lower atmosphere. In
addition, the poleward-moving disturbances may also be induced by the variation of the equatorial
electrojet [Chimonas, 1970] or just propagate from the southern hemisphere [Zakharenkova et al.,
2016]. A detailed investigation of this phenomenon is not the focus of this work.
**4. Discussion**
Our results show that the propagation parameters of the LSTID in the East Asian sector during
the St. Patrick's Day storm are longitudinal dependent. Among these parameters, the longitudinal
dependence of the propagation azimuth of an LSTID receives much attention in previous works. In
general, earlier studies suggest that there are four main factors that affect the direction of a polar
originated LSTID, which are the velocity of the background neutral wind [Hines, 1960; Morton and
Essex, 1978; Maeda and Handa, 1980], the structure and evolution of the source region in the auroral
oval [Maeda and Handa, 1980; Hunsucker, 1982; Ding et al., 2007], the Coriolis force [Maeda and
Handa, 1980; Balthazor and Moffett, 1999; Afraimovich et al., 2000; Tsugawa et al., 2004; Ding et
al., 2013], and the declination of geomagnetic field [Tsugawa et al., 2004; Jakowski et al., 2008;
Borries et al., 2009].
The Coriolis force effect is generally believed to contribute to the clockwise shift of the
propagation direction of the LSTIDs [Afraimovich et al., 2000; Tsugawa et al., 2004; Ding et al.,
2013], The observations of the shift (10°-20° on average) are consist with the calculation by Maeda
and Handa [1980] and the model simulation by Balthazor and Moffett [1999]. However, in our study,
the shift of the propagation direction is not systematic westward, which means the variability of the
LSTID azimuth in our observation cannot be attributed to the Coriolis force, at least not to it alone.
The structure/movement of the source region for the LSTID in the auroral oval is another
candidate for explaining the longitudinal dependence of the propagation direction of the LSTID.
Previous studies have suggested that the westward movement of enhanced electrojets in the auroral
arc is an important cause of the westward shift of the LSTID propagation direction at high latitudes
[Hunsucker, 1982; Ding et al., 2007]. The change of the propagation direction of LSTIDs as they
move from high to middle latitudes during the superstorm of 29 October 2003 over North America,
was explained by Ding et al. [2007] as related to a change in the position of the electrojet
enhancement area near the auroral oval. Nevertheless, since the structure and the evolution process
of the source region during storm period is complicated, more cases and modeling studies are needed
to find a clear connection between it and the propagation direction of LSTIDs.
In general, the velocity of the neutral wind is much less than that of the LSTIDs, and the
thermospheric wind velocity in the same latitudinal belt with a limited longitudinal extension should
exhibit little variance. So the contribution of the background wind on the change of the propagating
direction would be limited in the absence of the geomagnetic field. However, a combined effect of
magnetic declination and zonal wind can cause F region electron density differences between two
sides of the zero declination [Zhang et al., 2011]. During storm periods, the enhanced zonal winds
[Fuller-Rowell et al., 1994] can intensify these differences [Thomas et al, 2016]. As a result, the
geomagnetic declination is considered to be an important factor that affects the propagation
direction of the LSTID. Some researchers have studied the predominant propagation direction of





LSTIDs during storm periods in different longitudinal sectors, and suggest that, statistically speaking, the predominant directions of LSTID in European continent, China and Japan are primarily southward, South to West and South to East, respectively [Nicolls et al, 2004; Tsugawa et al, 2004; Jakowski et al, 2008; Borries 2009; Ding et al, 2013]. These results are all consistent with the corresponding geomagnetic declination in each sector.

In the longitudinal region of 70°E-150°E, the geomagnetic declination angles change from North by East in the West side to North by West in the East side. This characteristic seems to show some kind of consistent with the azimuth results in Table 1. To illustrate such connection quantitatively, Figure 9 depicts the (a) the geomagnetic declination on the wavefront in different longitudes in Figure 7 and (b) the propagation direction (azimuth-180°) of the LSTID at the same spot. The connection between these two parameters is quite obvious in this event. This result manifests that the propagation of LSTIDs in different longitudes is probably influenced by the orientation of the geomagnetic field lines in the East Asian sector. In addition, the tendency of field-aligned propagation of the LSTID indicates that it is driven by the neutral winds rather than by electric fields since the winds push the plasma up and down along the magnetic field lines. It should be noted that our speculation needs to be verified with more observational data and numerical simulation to reduce uncertainty in our propagation estimation and to figure out the detailed physical processes.

Zakharenkova et al. [2016] have studied the behaviors of LSTIDs during the St. Patrick's Day storm for the European and American sectors with GPS and GLONASS observations. It shows clearly in their results that the European sector (10°E) also exhibits LSTIDs around 11:00 UT. As a comparison, these LSTIDs are also investigated but with VTEC data from the Madrigal database of the MIT Haystack Observatory. This database has good spatiotemporal coverage for the European and American sectors. To derive the VTECP, a narrow longitudinal band (10°E-20°E, 30°N-70°N) is first selected and the VTEC data with the same latitude at the same time is average. At each latitude bin, the averaged VTEC forms a time series and the temporal resolution is reset to 12 minutes (0.2 hours) with bin averaging. Then, a running mean with a 1.5 hours window is conducted for each time series and their difference is taken as the VTECP. The result is plotted in Figure 10 as a TLP. The fitting lines are obtained with a similar method mentioned above.

Figure 10 is basically consistent with the Figure 3(b) in Zakharenkova et al. [2016], such as the synchronous perturbations around 04:45 UT and 09:15 UT, and the LSTID structures between 10:00 UT and 17:00 UT. Moreover, our result shows that the VTECP′ behavior between 60°N and 70°N is quite different from below. The pattern around 10:00 UT seems to represent a TID with smaller phase speed. Considering the physical processes are more complex in such high latitudes [Foster et al., 2014], we only focus on the perturbations below 60°N. The phase speeds estimated from the crest and trough are $\sim 500 \pm 51$ m/s and $\sim 427 \pm 55$ m/s, respectively, and the estimated period is $\sim 4.0 \pm 0.2$ hours. It is clear that the appearances of the LSTIDs are different in the European and the East Asian sectors during the same UT period for the same storm event. This may be related to the location or structure of the Joule heating source in the auroral oval or the difference of the background TEC in the two sectors. For better understanding this difference, more studies on the Joule heating source are needed.

**5. Summary**



Using data from 4 GPS receiver networks (CMGN, CMONOC, GEONET, IGS), together with
recordings of 2 HF Doppler shift stations and 8 ionosondes, we show the first observation results of
the LSTIDs in the East Asian sector during the 2015 St. Patrick's Day storm. The GPS receiver
networks in China and Japan are combined together to produce 2D VTEC perturbation maps in
order to give a wider image of the LSTID structures in the East Asia. As a comparison, the
ionospheric disturbances in the European sector are also studied with VTEC data from the Madrigal
database. The propagation parameters of the LSTIDs are estimated. Main results can be summarized
as follows:
(1) A negative LSTID occurs and propagates from high to lower latitudes during 09:40-11:20
UT, which spans over 60° in longitude. It is followed by a positive LSTID characterized by a clear
tendency to dissipate which starts from the East side. These features are in good agreement with
observations by HF Doppler shift stations and ionosondes
(2) The propagation orientation is almost due southward around 105°E-115°E, and it tends to
slightly shift westward/eastward in the West/East part of the studied area. This may be influenced
by the regional declination of the geomagnetic field lines.
(3) Other propagation parameters are also longitudinal dependent (see Table 1), and the mean
values and standard deviations of the period, Vt, Vc, and wavelength are $74.8\pm1.4$ minutes, $578\pm$
$16$ m/s, $617\pm23$ m/s, and $2691\pm80$ km, respectively.
**Acknowledgement:**
We are grateful to the International GPS Services (IGS) (ftp://cddis.gsfc.nasa.gov). The GPS data
from CMONOC and CMGN networks are provided by the China Earthquake Administration (CEA)
and the China Meteorological Administration (CMA), respectively. The GPS data from GEONET
are provided by the Geographical Survey Institute, Japan. GPS TEC data products and access
through the Madrigal distributed data system are provided to the community by the Massachusetts
Institute of Technology under support from the US National Science Foundation grant AGS-
1242204. The HF Doppler records are from the Chinese Meridian Project. The ionosonde data are
provided by the China Research Institute of Radio wave Propagation (CRIRP). We thank the
NASA/GSFC's Space Physics Data Facility's OMNIWeb service for data of the interplanetary and
SYM−H parameters. This research was supported by the National Natural Science Foundation of
China (No. 41674157).

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



**Captions of Table and Figures**
**Table 1.** The estimated propagation parameters of the LSTID and the corresponding standard errors.
The second column contains the propagation directions, which are measured clockwise from the
South. Vt/Vc represents the phase speed estimated with certain wave trough/crest.
**Figure 1.** Locations of the GPS stations of different networks (colored dots), the HF Doppler shift
stations (lime stars), the National Time Service Center of China (grey stars), and the ionosondes
(lime triangles) that used in this study.
**Figure 2.** Temporal variations of (a) the solar wind speed (Vsw), (b) the IMF Bz component, (c) the
SYM-H index, and (d) the AE index between 18:00 UT, 16 March 2015 and 06:00 UT, 18 March
2015. The occurrence of SSC is shown with vertical dashed lines.
**Figure 3.** Temporal variations of the HF Doppler shift records from (a) MDT and (b) SZT between
08:00 UT and 14:00 UT, 17 March 2015.
**Figure 4.** A series of 2D VTECP′ maps over the East Asian sector from the period of 09:40-09:50
UT to 11:30-11:40 UT on 17 March 2015. The grey areas represent the nightside. The colorbar
represents the VTECP′ (units: TECu). The lime and yellow lines illustrate the least square fittings
(order 2) for wavefronts.
**Figure 5.** Temporal variations of mean VTECP′ near the Doppler reflection points between 08:00
UT and 14:00 UT, 17 March 2015.
**Figure 6.** Temporal variations of the virtual height for iso-frequency lines from 8 ionosondes
between 08:00 UT and 12:00 UT, 17 March 2015. Frequencies are depicted on each iso-frequency
line. The time resolution is 15 min for all stations. The red dashed lines connect the peaks of the
highest and lowest frequencies.

**Figure 7.** A detailed example of the wavefront fitting method. Lime dots indicate the data points for
least square fitting. Lime arrows depict the propagation orientations in different longitudes. Dashed
black rectangles mark the areas for generating TLPs in Figure 8.

**Figure 8.** TLPs of VTECP′ for different longitudinal bands between 07:00-14:00 UT. White dots
give the data points for linear fitting, and the fitting results are marked with white lines. Black
dashed lines depict 30°N in (b-d, f) and 40°N in (f).

**Figure 9.** The sketch of (upper) the geomagnetic declination angels and (lower) the propagation
directions in different longitudes on the wavefront fitted in Figure 7. The propagation directions are
measured clockwise from the South.

**Figure 10.** The TLP of VTECP′ for the European sector (10°E-20°E, 30°N-70°N) between 01:00-
23:00 UT. White lines and dots are similar to those in Figure 8. The black dashed line depicts 60°N.



1   **Table 1.**

| Lon. (°E) | Dir. (°) | Period (min) | Vt (m/s) | Vc (m/s) | Wavelength (km) |
|---|---|---|---|---|---|
| 80-90 | -11.2 | 81.1±3.4 | 500±40 | 542±31 | 2536±163 |
| 90-100 | -7.1 | 77.6±5.2 | 552±22 | 670±44 | 2845±222 |
| 100-110 | -2.9 | 58.8±1.5 | 587±47 | 638±76 | 2160±167 |
| 110-120 | 1.3 | 62.4±2.0 | 605±27 | 562±25 | 2184±99 |
| 120-130 | 7.9 | 94.2±1.3 | 647±39 | 673±63 | 3731±216 |



1 Figure 1.

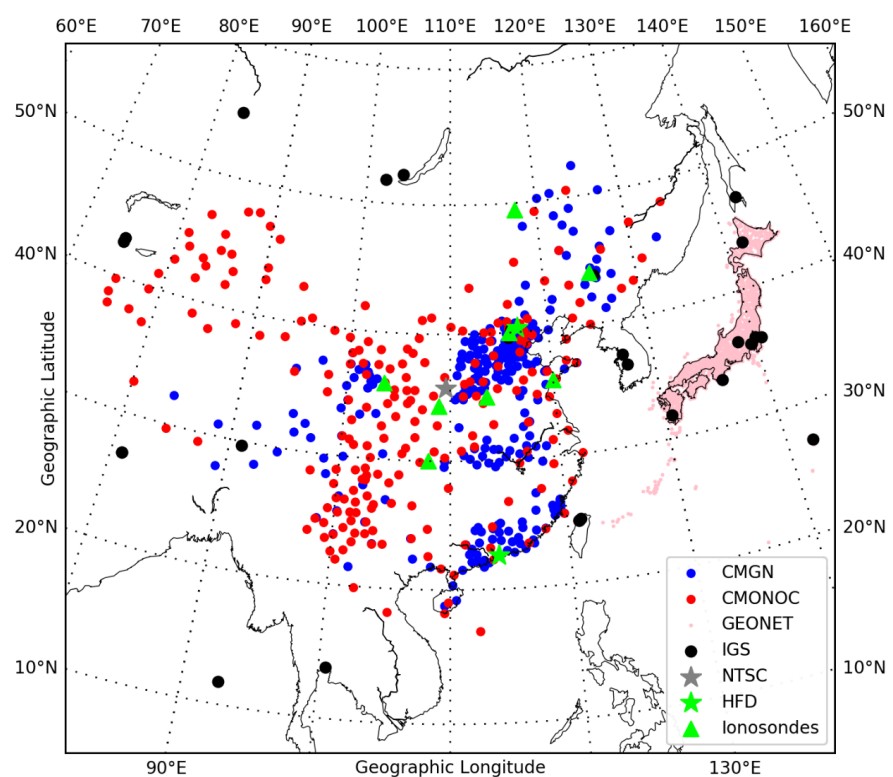



1    Figure 2.

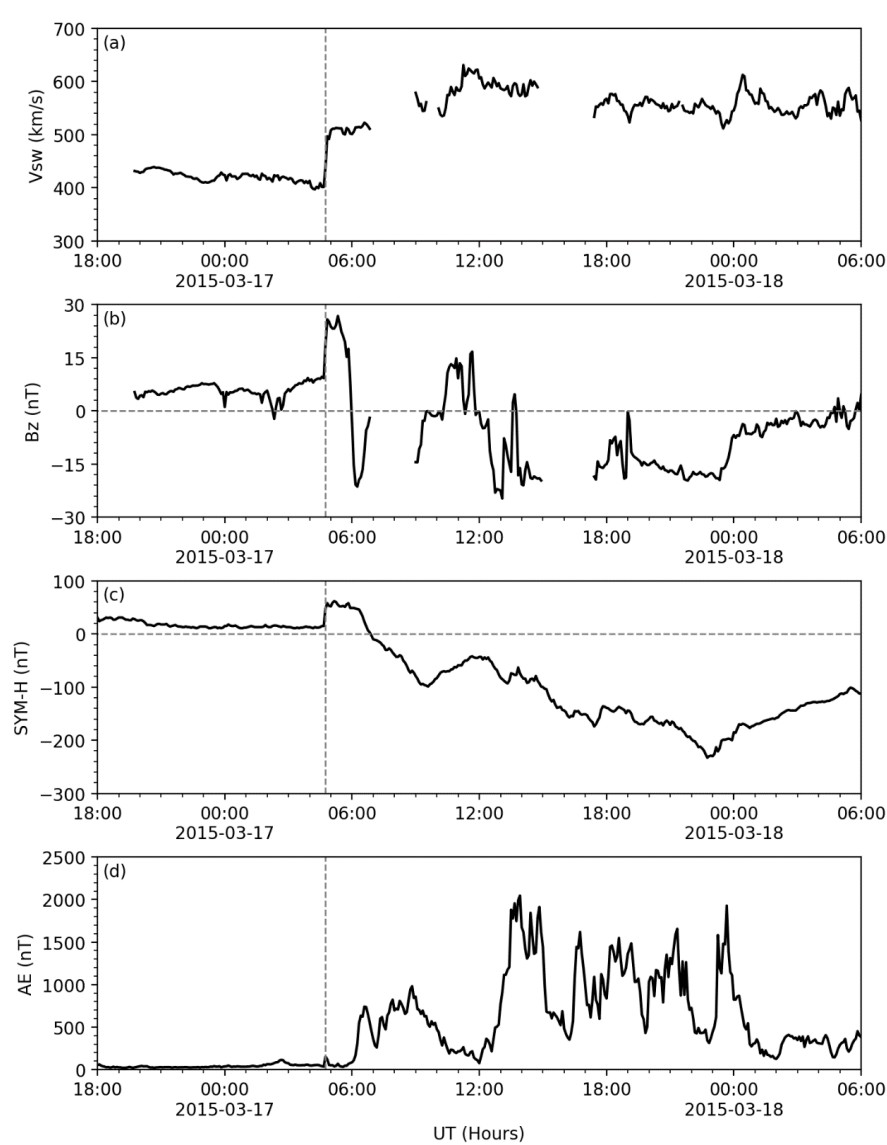

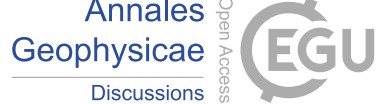



1    Figure 3.

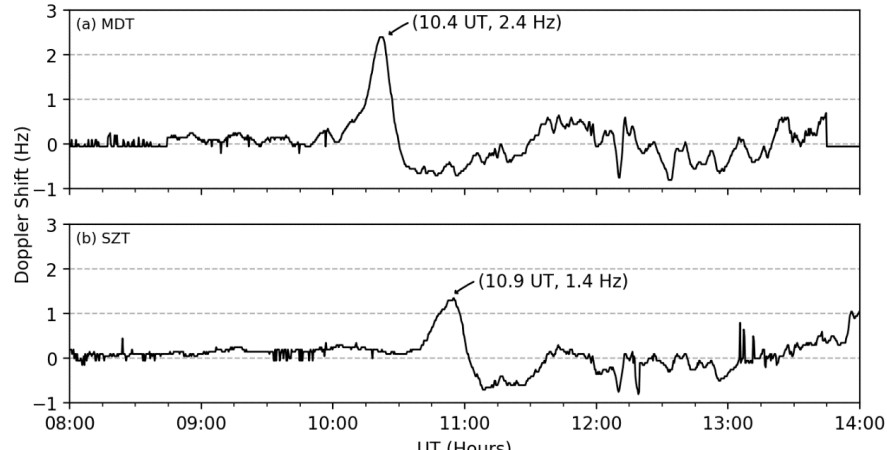

3



1    Figure 4.

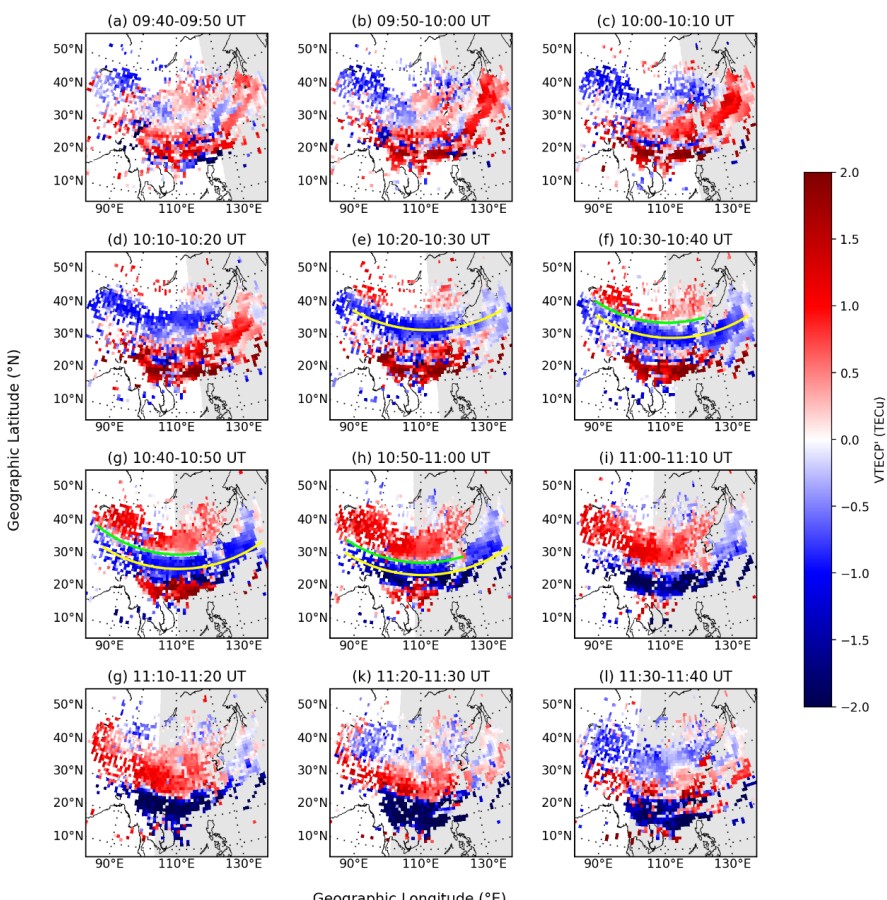



1    Figure 5.

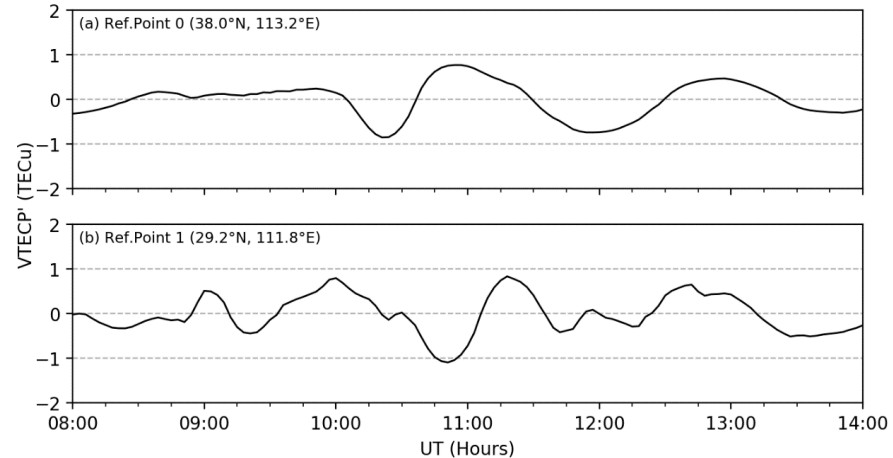



1    Figure 6.

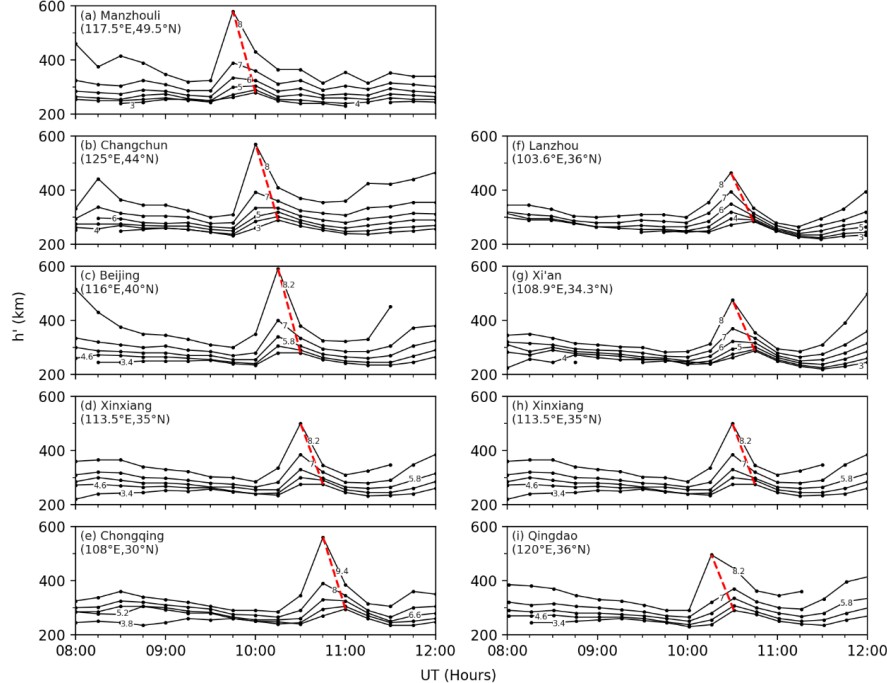



1    Figure 7.

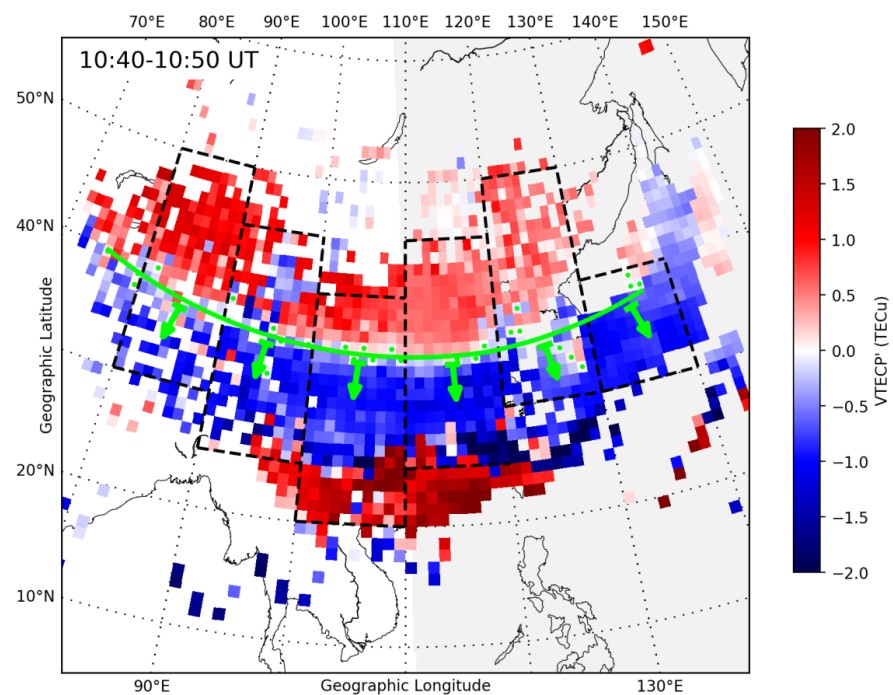

3





1    Figure 8.



1    Figure 9.

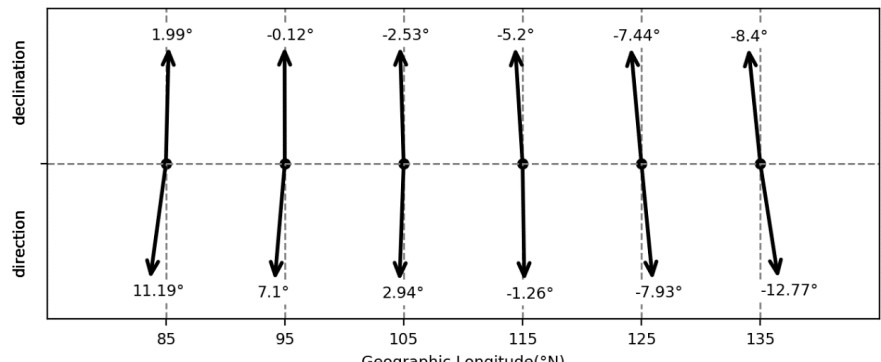

3





1    Figure 10.

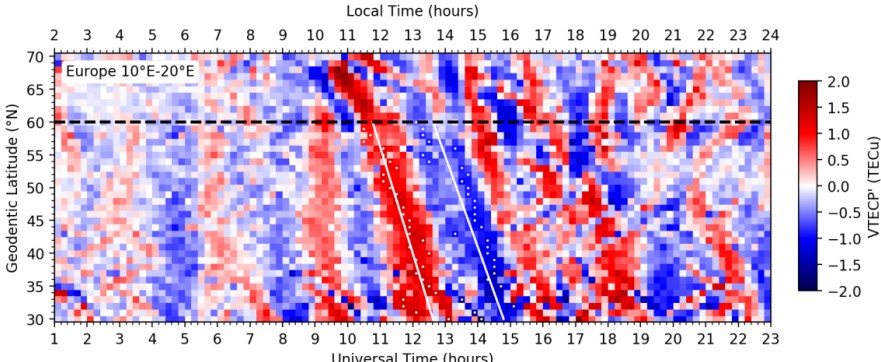

