# Peer review of "A case study of the large-scale traveling ionospheric disturbances in the East Asian sector during the 2015 St. Patrick's Day geomagnetic storm"

_Annales Geophysicae, 2019_

## Referee Comment (RC1) · Anonymous Referee #1 · 2 May 2019

**Review comments on manuscript *A case study of the large scale travelling ionospheric disturbances in the Eastern Asian sector during the 2015 St. Patrick's Day geomagnetic storm* by Liu et al (2019).**

02 May 2019

The current manuscript presents a detailed analysis of large scale travelling ionospheric disturbances which occurred during the March 17, 2015 geomagnetic storm. The authors combined data from multiple GNSS receiver networks to create 2-dimensional maps of total electron content perturbations from where azimuth, velocity and periods were determined. In addition, they made use of ionosondes and HF doppler radars, and in my view, have presented a very complete and comprehensive study especially focusing on the northern hemisphere mid-latitude region over the Asian region. They intentionally avoided discussing low latitude changes or TID related variations in this particular region (which I think they shouldn't have) as I will point out later in the comments. Nevertheless, this is an important contribution to the TIDs studies. In my view, the strength of this paper is two-fold

1. The utilisation of dense networks of diverse instrumentation (GNSS receivers, doppler radars and ionosondes) to bring out finer details of large scale TIDs during the March 17, 2015 geomagnetic storm.

2. The agreement of TID velocity estimated from 2-D TEC perturbation maps and HF doppler radar data. In fact the authors missed an opportunity to discuss this issue in detail and should be revised in the paper as it is an interesting one. Let me elaborate in more details here. From their Figure 3, they estimated the TID velocity to be 553 m/s (between 10:24-10:45 UT). Later in Figure 8, the estimated velocity from VTECP' maps (I assume it to be around the same time because the time of Vt and Cv was not actually clearly stated) was $578 \pm 16$m/s. Page 7, lines 36-37, they did comment that that '...Vt is in good agreement with the resultof 553 m/s derived from the Doppler observation'. Usually velocity values estimated from spaced instruments such as ionosondes and HF Doppler radars tend to be higher that the actual TIDs' velocities because it assumes "perfect equatorward propagation". If the spaced instruments (e.g., ionosondes or doppler radars are in 'perfect alignment' with the propagation of TIDs, then the velocities from the two methods can have a high degree of agreement. Taking a look at Figure 9 of the authors, their azimuth values point to this direction that the position of the HF dopper radars actually was in the direction of the TID's propagation. This may be the main reason why the velocity values from these two methods agree. I encourage the authors to add some discussion in this regard. Below are the references that talk directly to this second point

   - Afraimovich et al., (1998), GPS radio interferometry of travelling ionospheric disturbances, J. Atmos. Solar. Terr. Phys., 60, 1205-1223
   - Habarulema et al., (2013), Estimating the propagation characteristics of large-scale traveling ionospheric disturbances using ground-based and satellite data, J. Geophys. Res. Space Physics, 118, 7768-7782, doi:10.1002/2013JA018997

Below are detailed comments which may help the authors to improve their paper.

Abstract: Page 1, lines 13-15 which talk about first observation of LSTIDs in East Asian sector for the first time may not necessarily be entirely correct. I would like to point the authors to Habarulema et al., (2018)

which analysed LSTIDs during this storm period in the African, Asian and American regions. These authors reported LSTIDs in the Asian region between 0900-1200 UT reaching velocity values of over 800 m/s. Perhaps in the current paper, the authors have used more data and so their propagation parameters may be 'more accurate', but certainly this is not the first study over the Asian sector for this particular storm. Please re-word this accordingly.

*Reference:* **Habarulema et al., (2018)**: Storm time global observations of large-scale TIDs from ground-based and in situ satellite measurements. Journal of Geophysical Research: Space Physics, 123, 711-724, https://doi.org/10.1002/2017JA024510

Page 1, lines 17-19, the concept of negative and positive LSTID is not understandable. Usually TIDs are seen as periodic changes in VTECP' or electron density appearing like wave structures. As such, these structures would have 'troughs' and 'crests'. What are the authors calling negative LSTIDs?Is it where VTECP is negative? Shouldn't this be the 'troughs' of the wave or TID? Please check this and revise if necessary

Introduction: Page 2, lines 4-6, perhaps, the authors can talk about AGWs in general as they can also lead to MSTIDs? I think in the introduction, the authors missed critical papers which have done similar analysis for the March 17, 2015 storm. They include: **Borries et al., (2016)**: Multiple ionospheric perturbations during the Saint Patrick's Day storm 2015 in the European-African sector. Journal of Geophysical Research: Space Physics, 121, 11333-11345, https://doi.org/10.1002/2016JA023178; and **Ramsingh et al., (2016)**: Low-latitude ionosphere response to super geomagnetic storm of 17/18 March 2015: Results from a chain of ground-based observations over Indian sector. Journal of Geophysical Research: Space Physics, 120, 10864-10882, https://doi.org/10.1002/2015JA021509

Data and Methods: Page 4, lines 36-39 which discuss the ionosonde data that have been used in the study. Has this data been manually cross-checked to ensure that errornous ionograms were not used in the analysis and interpretation? This may be one of the errors associated with Figure 6 where the downward phase propagation did not manifest in a number of stations? I will comment on this later. For some reference about errors that could be in data due to wrongly scaling of ionograms, please see **Habarulema and Carelse (2016), I think their Figure 1?**: Long-term analysis between radio occultation and ionosonde peak electron density and height during geomagnetic storms, Geophys. Res. Lett., 43, 4106-4111, doi:10.1002/2016GL068944; and **Krankowski et al., (2011), Figures 4-5** : Ionospheric electron density observed by FORMOSAT-3/COSMIC over the European region and validated by ionosonde data, J. Geod., 85, 949-964

Observations: Page 5, lines 20-21, the text sounds misleading. The reader may go to Figure 1 looking for the 'disturbances which are observed successively' only to find the location of HF doppler radars.

Page 5, Line 23, the times indicated here are different from the times shown by arrows in Figure 3 of 10:24 UT and 10:45 UT?

Page 5, Lines 30-32, the authors should mention the limitation of this assumption to be valid when the AGW is in a perfectly equatorward direction and add references I mentioned in the opening statements

Page 5, Lines 33-34 are not clear. If the authors refer to the equation in section 2, then perhaps number it and refer to it here. Otherwise 'above' doesn't give appropriate guidance to the reader.

The motivation of transforming VTECP to VTECP' is not very clear to me. If the method used to estimate the background TEC values is consistently used, why would this be required? May be the VTECP values will be significantly small, but positive and negative perturbations should still come out of the 2-D maps?

Page 5, Line 39 and every where, Lime lines appear green? Not sure of the color. The issue which

requires more details is where the authors talk about 'values close to zero'. How close to zero? Is it possible to provide a range may be between -0.05 and 0.05?

Page 6, Lines 10-17, I suggest that the author have a look at the paper by **Pradipita et al., (2016)**: Interhemispheric propagation and interactions of auroral traveling ionospheric disturbances near the equator. Journal of Geophysical Research: Space Physics, 121, 2462-2474, https://doi.org/10.1002/2015JA022043

Page 6, Lines 18-27, considering Figure 5 where there is a temporal shift seen at 38 degrees North followed by 29 degrees North, have the authors considered investigating such LSTID to be originating from northern hemisphere and propagation towards the equator with possibility of crossing the equator towards the southern hemisphere?

Page 6, lines 28-44, In my view, ionosonde data and its interpretation should be given more attention than is done in the current version. If the uplift of virtual height (h') is due to the AGW which results into the TID that reaches ionospheric heights, then we would have seen the dominant trend in downward phase velocity. An important consideration with ionosonde data is to check that scaling was done correctly as I have already mentioned. In the current manuscript, the authors paid too much attention to the higher iso-line and connected it to the one at the lower h'. In my opinion, this should be re-looked at, because the ionosonde is 'more accurate' at measuring the bottomside ionosphere. Although I don't know the actual height corresponding to virtual height of 600 km, it may be possible that this could fall within the extrapolated topside? Therefore, in actual sense, we should be able to see the downward phase velocities from the lower h' values for the analysis to be reliable. For a recent analysis of ionosonde data during this storm period, please see `https://doi.org/10.1002/2017JA024510`.

Estimating Propagation Parameters: I would like to first commend the authors for presenting a straight forward and easy to understand method of doing this.

Page 7, lines 11-12, the text which talks about morphology changes of the TID changing as it propagates from high to lower latitudes: Is this backed by any references? Because the analysis of the authors is limited to northern hemisphere mid-latitudes

Page 7, lines 15-20: The authors can consider labelling the 'rectangles' as A, B, C, etc or something along this line for the reader to easily identify them in the Figure.

Page 7, lines 37-38: This is where a discussion/description of the agreement between the two techniques (VTECP' and doppler radar) should have discussed. Please refer to my earlier comment in the opening statements.

Page 8, line 6, after Chimonas , 1970; add a reference `https://doi.org/10.1002/2016GL069740` as these authors directly reported related results based on GNSS TEC observations and other measurements. Very recently, **Jonah et al., (2018)**, available on `https://doi.org/10.1029/2018JA025367`, reported related results during storm conditions. Consult this reference as well (I think their Figure 4). On this point, `https://doi.org/10.1002/2017JA024510` reported equatorward LSTID propagating from the southern hemisphere crossing to the northern hemisphere in the Asian region during this storm period. In fact, their analysis showed that these TIDs may not have exceeded 30 degrees North, which may be in agreement with your analyses and is more clearer in Figure 8(c) at around 1200 UT. Please have a look at their Figure 3(e) and possibly add some discussion to this effect.

Summary: Page 10, lines 1-3: The authors may want to rephrase this statement given that an earlier study by Habarulema et al., (2018)– `https://doi.org/10.1002/2017JA024510`– provided some analysis for this particular storm in the Asian sector. May be the analyses was not as detailed as provided in this paper, but definately this is not the first analysis for this storm in the Asian region. The strength of this paper over what was presented in Habarulema et al., (2018) and other attempted studies is the use of multiplicity of data sources to provide more details and clarity during this storm period.

---

## Referee Comment (RC2) · Anonymous Referee #2 · 26 May 2019

The manuscript addresses the interesting scientific problem of understanding the properties of large scale travelling ionospheric disturbances (LSTIDs), which are frequently observed during geomagnetic storms. This manuscript discusses the properties of LSTIDs during 17th March 2015 with focus on the Chinese and Japanese sector. Although, descriptions of LSTID occurrence during this event have been published before, this manuscript adds new aspects on the longitudinal dependence of the LSTID properties in the Chinese/ Japanese sector based on GNSS, HF and ionosonde data. The manuscript is well structured, well written and presents analysis of high quality in a well understandable way. Thus, my overall evaluation is publishing after solving minor remarks. The manuscript in its current form has three weak points.

I. First, already in the abstract the authors are referring to negative and positive LSTIDs and seem to treat them in the course of the manuscript like separate phenomenon. Since these LSTIDs are the signature of atmospheric gravity waves, both signatures belong to the same wave. Therefore, I would recommend to avoid discussing positive and negative amplitudes separately.

II. Second, the key point of the manuscript is the discussion of longitudinal dependence of LSTID properties. But, this is impacted by the data coverage. The data coverage is lower in the east and west boundaries of the investigated region. I argue that this impacts the accuracy of the estimation of the LSTID properties. The discussion of the LSTID properties (wavelength, period and speed) should be treated with more care concerning reliability of the results.

III. Third, the authors present also the result of the LSTID occurrence in Europe. This has been extensively discussed in Borries et al. (2016, https://doi.org/10.1002/2016JA023178 ). Specifically the LSTID occurring between 11 and 12 UT has been discussed to be "special" because it is impacted by winds and prompt penetration electric fields at the same time. This fact should be included in the discussion of this manuscript. It supports the finding of the authors that the LSTID properties in Europe differ from the LSTID properties in the Chinese/ Japanese sector.

Next, I like to add some technical recommendations/corrections:

1. Jakowski et al. (2008) did not discuss LSTIDs but large scale gradients (no discussion of wave properties)

2. URLs of the data sources are usually provided in the acknowledgements, not in the text.

[Figure]

3. Presenting the LSTID results with VTECP' has the advantage of better illustrating most wave properties, but it does not represent the true wave amplitude anymore. On a quick view, the figures might be misinterpreted. Therefore, I recommend to make it very clear that this is an "artificial" amplitude.

4. Figures 3 and 5 do not have much content. But they are supposed to be compared with each other. Therefore, I recommend to join the content of both figures into one figure. This will increase the information density and allow better comparability.

5. In figure 8, the impact of EIA is addressed. I assume, the dashed black lines indicate the boundary of EIA. This should be made clear in the text and figure description.

6. In the discussion section, the authors exclude the impact of electric fields on the LSTID propagation and favour the impact of winds, driving the LSTID propagation because of field-aligned propagation. For a better understanding, the authors should explain, what would be different in case of electric field impact. In fact, since Borries et al. (2016) describe prompt penetration electric field impact in Europe at that time, more emphasis should be given to discuss electric field impact in the Chinese/Japanese sector at the same time.

7. I detected a few spelling errors and grammar issues (indicated in the supplementary material). I expect, there are more than I found and recommend professional editing.

Please also note the supplement to this comment:
https://www.ann-geophys-discuss.net/angeo-2019-63/angeo-2019-63-RC2-supplement.pdf

[Figure]

**Supplement:**

**A case study of the large-scale traveling ionospheric disturbances in the East Asian sector during the 2015 St. Patrick's Day geomagnetic storm**

J. Liu[1], D. H. Zhang[1]*, A. Coster[2], S. R. Zhang[2], G. Y. Ma[3], Y. Q. Hao[1], Z. 
[revised manuscript text omitted]

[Figure]

1    Figure 2.

[Figure]

1    Figure 3.

[Figure]

3

1    Figure 4.

[Figure]

1    Figure 5.

[Figure]

1    Figure 6.

[Figure]

1    Figure 7.

[Figure]

3

1    Figure 8.

[Figure]

1 Figure 9.

[Figure]

3

1    Figure 10.

[Figure]

---

## Author Comment (AC1) · 4 Jun 2019

Thank you for your substantial and detailed comments! According to these comment, we revised the manuscripts and gave our replies to these comments point by point. The file of response to the comments and the revised manuscript with annotations to these comment are provided in the supplement. By the way, the reviewer #2 comments were also provided last week, but we havn't commpleted the response and revision according to reviewer #2 comments. So, this manuscript is just revised according to reviewer #1 comments.

Thank you very much!

[Figure]

Please also note the supplement to this comment:
https://www.ann-geophys-discuss.net/angeo-2019-63/angeo-2019-63-AC1-supplement.zip

---

## Author Comment (AC2) · 12 Jun 2019

Thank you for your substantial and detailed comments here and in the supplement material! According to these comments and corresponding annotations in the supplement material, we revised the manuscripts on the basis of the last revised draft based on reviewer 1 comments. And gave our replies point by point. The red colors mark the revised parts and the new references. Since this revision is based on the revised manuscript according to Review Comments #1. The revised part related to reviewer 1 comments is marked with yellow highlights.

the final revised manuscript and detailed response letter are provided in supplement
material

Please also note the supplement to this comment:
https://www.ann-geophys-discuss.net/angeo-2019-63/angeo-2019-63-AC2-supplement.zip

———————————————————

---

## Author Response (AR1)

**Reply to the Review Comments #1 and #2**

**Contents**

**Section 1. Reply to the Review Comments #1.**

**Section 2. Reply to the Review Comments #2.**

**Section 3. Marked-up manuscript. Pages 1-28.**

Note: The **bolded Page Numbers** in Replies of Section 1&2 refer to page numbers of Section 3

**Reply to the Review Comments #1**

Thank you for your substantial and detailed comments! According to these comments, we revised the manuscripts and gave our replies to these comments point by point. The original manuscript, the revised version and the added references are listed if necessary. The yellow highlights mark the revised parts and the new references according to the Review Comments #1.

Note that this reply is focusing on the Review Comments #1. The revised parts and the new references according to the Review Comments #2 are marked with red colors when there are overlaps.

**Major comments:**

**The current manuscript presents a detailed analysis of large scale travelling ionospheric disturbances which occurred during the March 17, 2015 geomagnetic storm. The authors combined data from multiple GNSS receiver networks to create 2-dimensional maps of total electron content perturbations from where azimuth, velocity and periods were determined. In addition, they made use of ionosondes and HF doppler radars, and in my view, have presented a very complete and comprehensive study especially focusing on the northern hemisphere mid-latitude region over the Asian region. They intentionally avoided discussing low latitude changes or TID related variations in this particular region (which I think they shouldn't have) as I will point out later in the comments. Nevertheless, this is an important contribution to the TIDs studies. In my view, the strength of this paper is two-fold**

**1. The utilization of dense networks of diverse instrumentation (GNSS receivers, doppler radars and ionosondes) to bring out finer details of large scale TIDs during the March 17, 2015 geomagnetic storm.**

**2. The agreement of TID velocity estimated from 2-D TEC perturbation maps and HF doppler radar data. In fact the authors missed an opportunity to discuss this issue in detail and should be revised in the paper as it is an interesting one. Let me elaborate in more details here. From their Figure 3, they estimated the TID velocity to be 553 m/s (between 10:24-10:45 UT). Later in Figure 8, the estimated velocity from VTECP' maps (I assume it to be around the same time because the time of $V_t$ and $V_c$ was not actually clearly stated) was 578±16m/s. Page 7, lines 36-37, they did comment that that '...$V_t$ is in good agreement with the result of 553 m/s derived from the Doppler observation'. Usually velocity values estimated from spaced instruments such as ionosondes and HF Doppler radars tend to be higher that the actual TIDs' velocities because it assumes "perfect equatorward propagation". If the spaced instruments (e.g., ionosondes or doppler radars are in 'perfect alignment' with the propagation of TIDs, then the velocities from the two methods can have a high degree of agreement. Taking a look at Figure 9 of the authors, their azimuth values point to this direction that the position of the HF Doppler radars actually was in the direction of the TID's propagation. This may be the main reason why the velocity values from these two methods agree. I encourage**

**the authors to add some discussion in this regard. Below are the references that talk directly to this second point**

• **Afraimovich et al., (1998), GPS radio interferometry of travelling ionospheric disturbances, J. Atmos. Solar. Terr. Phys., 60, 1205-1223**

• **Habarulema et al., (2013), Estimating the propagation characteristics of large-scale traveling ionospheric disturbances using ground-based and satellite data, J. Geophys. Res. Space Physics, 118, 7768-7782, doi:10.1002/2013JA018997**

**Reply to Major Comment:**

Thanks for your evaluation of this manuscript and good suggestions. Based on your suggestion, we try to add some discussions about the result of HF Doppler and VTECP'. Considering this suggestion and the detail comments, the following paragraph about the relationship of the velocity of LSTID derived from HF Doppler and VTECP' is added in the revised manuscript.

**Page 8, Lines 34-43; Page 9, Line 1**:"In addition, it is interesting to note that Vt is in reasonable agreement with the result of 535 m/s derived from the Doppler recordings. To show it more specifically, we estimated the speed and direction of the LSTID using the same TLP method as Figure 8 but in 111°E-114°E and 29°N-38°N (corresponding to the reflecting points). The result is 562±59 m/s and 0°, respectively. In general, the LSTID velocity estimated from ground-based stations tend to be larger than the actual velocity since these stations, in most cases, are not in perfect alignment with the propagation direction of the LSTID [Afraimovich et al., 1998; Habarulema et al., 2013]. Such good agreement between VTECP' and HF Doppler results may be attributed to the fact that the reflecting points (29.2°N,111.8°E; 38.0°N,113.2°E) of the Doppler receivers are in a narrow longitudinal band and the direction of the LSTID's propagation is also almost due south between 111°E-114°E."

**Detail comments**

**Abstract:**

**Comment 1**

Page 1, lines 13-15 which talk about first observation of LSTIDs in East Asian sector for the first time may not necessarily be entirely correct. I would like to point the authors to Habarulema et al., (2018) which analysed LSTIDs during this storm period in the African, Asian and American regions. These authors reported LSTIDs in the Asian region between 0900-1200 UT reaching velocity values of over 800 m/s. Perhaps in the current paper, the authors have used more data and so their propagation parameters maybe 'more accurate', but certainly this is not the first study over the Asian sector for this particular storm. Please re-word this accordingly.

Reference: Habarulema et al., (2018): Storm time global observations of large-scale TIDs from ground-based and in situ satellite measurements. Journal of Geophysical Research: Space Physics, 123, 711-724, https://doi.org/10.1002/2017JA024510.

**Section 1. Reply to the Review Comments #1**

**Reply 1**

Thanks for pointing it out. We missed this reference and made the inappropriate statement of the "first observation". According to this comment, the manuscript is revised. The reference "Habarulema et al., (2018)" is added in the reference list, and necessary discussions are also given in the revised manuscript (e.g., Revision 3, 17).

The statements in the original manuscript

"This study gives the first observation of the large-scale traveling ionospheric disturbances (LSTIDs) in the East Asian sector during the 2015 St. Patrick's Day (March 17, 2015) geomagnetic storm.

…

Using data from 4 GPS receiver networks (CMGN, CMONOC, GEONET, IGS), together with recordings of 2 HF Doppler shift stations and 8 ionosondes, we show the first observation results of the LSTIDs in the East Asian sector during the 2015 St. Patrick's Day storm."

are revised as:

**Page 1, Lines 10-12:** "This study presents a comprehensive observation of the large-scale traveling ionospheric disturbances (LSTIDs) in the East Asian sector during the 2015 St. Patrick's Day (March 17, 2015) geomagnetic storm."

…

**Page 11, lines 25-28:** "Using data from 4 GPS receiver networks (CMGN, CMONOC, GEONET, IGS), together with recordings of 2 HF Doppler shift stations and 8 ionosondes, we provide comprehensive and detailed observation results of the LSTIDs in the East Asian sector during the 2015 St. Patrick's Day storm."

**Added Reference**

**Page 14, lines 27-29:** Habarulema, J. B., Yizengaw, E., Katamzi-Joseph, Z. T., Moldwin, M. B., and Buchert, S.: Storm Time Global Observations of Large-Scale TIDs From Ground-Based and In Situ Satellite Measurements. Journal of Geophysical Research: Space Physics, 123(1), 711-724, 2018.

**Comment 2**

Page 1, lines 17-19, the concept of negative and positive LSTID is not understandable. Usually TIDs are seen as periodic changes in VTECP' or electron density appearing like wave structures. As such, these structures would have 'troughs' and 'crests'. What are the authors calling negative LSTIDs? Is it where VTECP is negative? Shouldn't this be the 'troughs' of the wave or TID? Please check this and revise if necessary.

**Reply 2**

Very thanks! According to this comment, the some statements in the original manuscript are revised.

"Results show that a negative LSTID spanning at least 60° in longitude (80°E-140°E) occurs and propagating from high to lower latitudes around 09:40-11:20 UT. It is followed by a positive LSTID which shows a tendency of dissipation starting from the East side."

is revised as:

**Section 1. Reply to the Review Comments #1**

**Page 1, Lines 14-17:** "Results show that a trough of LSTID spanning at least 60° in longitude (80°E-140°E) occurs and propagates from high to lower latitudes around 09:40-11:20 UT. It is followed by a crest of LSTID which shows a tendency of dissipation starting from the East side.

"(1) A negative LSTID occurs and propagates from high to lower latitudes during 09:40-11:20 UT, which spans over 60° in longitude. It is followed by a positive LSTID characterized by a clear tendency to dissipate which starts from the East side."

is revised as:

**Page 11, lines 33-35:** "(1) A trough of LSTID occurs and propagates from high to lower latitudes during 09:40-11:20 UT, which spans over 60° in longitude. It is followed by a crest of LSTID that characterized by a clear tendency to dissipate starting from the East side."

**Introduction:**

**Comment 3**

Page 2, lines 4-6, perhaps, the authors can talk about AGWs in general as they can also lead to MSTIDs? I think in the introduction, the authors missed critical papers which have done similar analysis for the March 17, 2015 storm. They include: Borries et al., (2016): Multiple ionospheric perturbations during the Saint Patrick's Day storm 2015 in the European-African sector. Journal of Geophysical Research: Space Physics, 121, 11333-11345, https://doi.org/10.1002/2016JA023178; and Ramsingh et al., (2016): Low-latitude ionosphere response to super geomagnetic storm of 17/18 March 2015: Results from a chain of ground-based observations over Indian sector. Journal of Geophysical Research: Space Physics, 120, 10864-10882, https://doi.org/10.1002/2015JA021509

**Reply 3**

According to this comment, the first paragraph is rewritten and some references are cited in suitable place in the revised manuscript.

**The first and last paragraphs of Introduction**

"In general, the response of the ionosphere to the geomagnetic storm is classified by a variety of different features, one of which is the large scale traveling ionospheric disturbance (LSTID). It is the wave-like perturbation mainly propagating equatorward from high latitudes that is considered to be the manifestation of the presence of atmospheric gravity waves (AGWs) within the ionosphere caused by Joule heating or Lorenz-drag forcing in the Auroral regions during geomagnetic storm period [Hines, 1960; Richmond and Roble, 1979; Hocke and Schlegel, 1996].

......

During the period of 17–18, March 2015, a large geomagnetic storm occurred, which is the strongest one in the 24th solar cycle. During this storm, LSTIDs over European and American sectors are detected and analyzed with data from Global Navigation Satellite Systems (GNSS) stations [Zakharenkova et al., 2016]. Meanwhile, two high frequency (HF) Doppler stations operated by China Meridional Project [Wang, 2010] at mid-latitude China record a large ionospheric HF Doppler shift after 10:00 UT, which appears to indicate LSTIDs related to the geomagnetic storm. In this study, the multi-network of densely distributed GPS receivers and an ionosonde network will be used to reveal the propagating characteristics of these large disturbances recorded in HF Doppler shift receivers in the East Asian region, especially the characteristics of the dominant propagating direction of this disturbance over China and Japan."

are revised as:

  **Page 2, lines 4-12:** "In general, the response of the ionosphere to the geomagnetic storm is classified by a variety of different features, one of which is the large scale traveling ionospheric disturbance (LSTID) that is the wave-like perturbation mainly propagating equatorward from high latitudes. Traveling ionospheric disturbances (TIDs) are classified into LSTIDs and Medium-scale TIDs and they are considered to be the ionospheric manifestation of the presence of atmospheric gravity waves (AGWs) stimulated by different sources. LSTIDs are mainly caused by Joule heating or Lorenz-drag forcing in the Auroral regions during geomagnetic storm period [Hines, 1960; Richmond and Roble, 1979; Hocke and Schlegel, 1996]."

  ……

  **Page 3, lines 24-44:** "During the period of 17–18, March 2015, the strongest geomagnetic storm in the 24th solar cycle occurred and LSTIDs are detected and analysed in different longitudinal sectors [Ramsingh et al., 2015; Borries et al., 2016; Zakharenkova et al., 2016; Habarulema et al, 2018]. Meanwhile, two high frequency (HF) Doppler stations operated by China Meridional Project [Wang, 2010] at mid-latitude China record large ionospheric HF Doppler shifts after 10:00 UT, which seem to indicate the LSTIDs in the Asian region between 09:00-12:00 UT that reported by Habarulema et al. [2018]. In this study, the multi-network of densely distributed GPS receivers, the HF Doppler stations, and an ionosonde network are used to conduct a more comprehensive study on the propagating characteristics of the disturbances in the East Asian region, especially on the characteristics of the dominant propagating direction over China and Japan."

**Following References are added in the reference list**

**Page 13, lines 16-18:** Borries, C., Mahrous, A. M., Ellahouny, N. M., and Badeke, R.: Multiple ionospheric perturbations during the Saint Patrick's Day storm 2015 in the European-African sector. Journal of Geophysical Research: Space Physics, 121(11), 11-333, 2016.

**Page 15, lines 32-35:** Ramsingh, Sripathi, S., Sreekumar, S., Banola, S., Emperumal, K., Tiwari, P., and Kumar, B. S.: Low-latitude ionosphere response to super geomagnetic storm of 17/18 March 2015: Results from a chain of ground-based observations over Indian sector. Journal of Geophysical Research: Space Physics, 120(12), 10-864, 2015.

**Data and Methods:**

**Comment 4**

  Page 4, lines 36-39 which discuss the ionosonde data that have been used in the study. Has this data been manually cross-checked to ensure that errornous ionograms were not used in the analysis and interpretation? This may be one of the errors associated with Figure 6 where the downward phase propagation did not manifest in a number of stations? I will comment on this later. For some reference about errors that could be in data due to wrongly scaling of ionograms, please see Habarulema and Carelse (2016), I think their Figure 1?: Long-term analysis between radio occultation and ionosonde peak electron density and height during geomagnetic storms, Geophys. Res. Lett., 43, 4106-4111, doi:10.1002/2016GL068944; and Krankowski et al., (2011), Figures 4-5 : Ionospheric electron density observed by FORMOSAT-3/COSMIC over the European region and validated by ionosonde data, J. Geod., 85, 949-964

**Reply 4**

Thank you for raising concerns about the error checking.

The virtual height data in Figure 6 is manually scaled by ourselves from ionograms with professional scaling software provided by the China Research Institute of Radio wave Propagation (CRIRP). The validity of these ionograms is supported by this institute and our manually scaling has been checked by ourselves for a few times. We will make this clear in the revision.

We tend to believe that the data is not wrong. But as the reviewer mentioned, the downward phase propagation is not clear in a number of stations. We think this may be attributed to the relatively low temporal resolution of 15 minutes.

**According to this comment, we revised the statement**"In this study, ionograms from 8 ionosonde stations in China middle latitude are used to derive the iso-frequency lines, which vary as a function of universal time and virtual height. The sample rate of the ionograms is 15 minutes. These ionosondes belong to the China Research Institute of Radio-wave Propagation (CRIRP) and their locations are marked in Figure 1 with lime triangles. In addition, the condition of the geomagnetic storm is shown with data from the high resolution (5 minutes) OMNI dataset, which is downloaded from the FTP service of the NASA Goddard Space Flight Center (https://spdf.gsfc.nasa.gov)." as

**Page 4, Line 39 ~ Page 5, Lines 1-9:** "In this study, ionograms from 8 ionosonde stations in China middle latitude are used to derive the iso-frequency lines, which vary as a function of universal time and virtual height. The sample rate of the ionograms is 15 minutes. These ionosondes belong to the China Research Institute of Radio-wave Propagation (CRIRP) and their locations are marked in Figure 1 with green triangles. The virtual height data is manually scaled by ourselves to reduce possible errors of auto scaling [Krankowski et al., 2011; Habarulema and Carelse, 2016] from these ionograms with professional scaling software provided by CRIRP. During the scaling, we limited the frequency to be less than 7 MHz. In addition, the condition of the geomagnetic storm is shown with data from the high resolution (5 minutes) OMNI dataset, which is downloaded from the FTP service of the NASA Goddard Space Flight Center."

**And the two references are added in the reference list.**
**Page 14, Lines 18-20:** Habarulema, J. B. and Carelse, S. A.: Long-term analysis between radio occultation and ionosonde peak electron density and height during geomagnetic storms. Geophysical Research Letters, 43(9), 4106-4111, 2016.
**Page 15, Lines 6-8:** Krankowski, A., Zakharenkova, I., Krypiak-Gregorczyk, A., Shagimuratov, I. I., and Wielgosz, P.: Ionospheric electron density observed by FORMOSAT-3/COSMIC over the European region and validated by ionosonde data. Journal of Geodesy, 85(12), 949-964, 2011.

**Observations:**

**Comment 5**

Page 5, lines 20-21, the text sounds misleading. The reader may go to Figure 1 looking for
the 'disturbances which are observed successively' only to find the location of HF doppler radars.
**Reply 5**
Thank you for pointing it. The misleading expression "During the first stage of the main
phase, disturbances are observed successively at two HF Doppler receiver stations shown in
Figure 1"
is revised as:

**Page 5, Lines 30-31:** "During the first stage of the main phase, disturbances are observed
successively at MDT and SZT Doppler receiver stations."

**Comment 6**
Page 5, Line 23, the times indicated here are different from the times shown by arrows in
Figure 3 of 10:24 UT and 10:45 UT?

**Reply 6**
Thank you for pointing out this discrepancy. After checking the data we find that the
occurrence time of the positive shifts are about 10:22 UT and 10:53 UT. The speed estimated from
HF Doppler is about 535 m/s.
According to this, figure 3 is redrawn and the statement
"It shows that two distinct positive shifts occur at about 10:21 UT and 10:55 UT,
respectively. … the approximate speed of this perturbation is about 553 m/s."
is revised as:

**Page 5, Lines 32-33, 36-37:** "It shows that two distinct positive shifts occur at about 10:22
UT and 10:53 UT, respectively. … the approximate speed of this perturbation is about 535 m/s."

**Revised figure 3**

[Figure]

**Comment 7**

Page 5, Lines 30-32, the authors should mention the limitation of this assumption to be valid when the AGW is in a perfectly equatorward direction and add references I mentioned in the opening statements

**Reply 7**

Thank you for the thought-provoking suggestion.

With this sentence, we just want to give a rough classification of the disturbances based on the magnitude of the estimated speed. The detail illustration and velocity estimation will be given in the following text with VTECP method. We have revised the text to make it clear.

**Please refer to Reply 16 for detail. The references are also added in Revision 16.**

The sentence in the original manuscript "Considering the propagating speed and the interval of the positive-negative variations, the recorded perturbations probably reflect an equatorward propagating LSTID in the East Asian sector."

is revised as:

**Page 5, lines 40-42:** "Considering the magnitude of the speed and the time interval of the positive-negative variations, the recorded perturbations probably reflect an equatorward propagating LSTID in the East Asian sector."

**Comment 8**

Page 5, Lines 33-34 are not clear. If the authors refer to the equation in section 2, then perhaps number it and refer to it here. Otherwise 'above' doesn't give appropriate guidance to the reader.

**Reply 8**

Thank you very much!

With "above" we want to refer to the VTECP map method described in section 2 in the paragraph that includes the equations but not just the equations. As the comment describes, just say "above" is inappropriate indeed and it is reworded to "in section 2". The equations are also numbered.

**According to this comment, the sentence is revised as:**

**Page 6, lines 1-2:** "To confirm this, Figure 4 presents a sequence of 2D VTECP maps between 09:40-11:40 UT on 17 March 2015 with the method described in section 2."

**Comment 9**

The motivation of transforming VTECP to VTECP' is not very clear to me. If the method used to estimate the background TEC values is consistently used, why would this be required? May be the VTECP values will be significantly small, but positive and negative perturbations should still come out of the 2-D maps?

**Reply 9**

Thank you very much for pointing out the confusing expression.

In short, this transformation is not absolutely necessary but it will make the colormap of the
2D VTECP plots looks better. Below is the original 2D VTECP map without the transform.

[Figure]

The fact is that the perturbations in the EIA region is much larger than that in the mid
latitudes. If we choose a colormap based on the values of the lower latitudes, the color in the mid
latitudes will be too light (left figure). If we choose a colormap based on the values of the mid
latitudes, the color in the lower latitude will be too dark and lose details (right figure).

[Figure]

The left is the plot of the transform equation $y = sgn(x) * log_{10}(abs(x) + 1)$. It
accelerates the changes when x approaches 0 and decelerates the changes when x deviates 0. So
with this transform, we can sharpen the edges between positive and negative values and reduce the
differences of large and small absolute values. The outcome is the right figure.
We have revised the text to make this clear.
**The original description**
"Note that the raw value of VTECP is converted into VTECP′ with
$$VTECP' = sign(VTECP) * log_{10}(abs(VTECP) + 1)$$
to make it easier to distinguish the regions with positive and negative perturbations."
**is revised as:**

**Page 6, Lines 2-7:** "The raw value of VTECP has already been converted into VTECP′ with
the equation

$$VTECP' = sgn(VTECP) * log_{10}(abs(VTECP) + 1) \qquad (3)$$

The raw amplitude of VTECP above 30°N is ~ 2 TECu while the raw amplitude of VTECP below
30°N reaches ~ 10 TECu. So, transform (3) provides a better colormap for 2D VTECP plots by sharpening the edges between positive and negative values and reduce the differences of VTECP in middle and low latitudes."

**Comment 10**

Page 5, Line 39 and everywhere, Lime lines appear green? Not sure of the color. The issue which requires more details is where the authors talk about 'values close to zero'. How close to zero? Is it possible to provide a range may be between -0.05 and 0.05?

**Reply 10**

Thank you very much for pointing out the confusing parts.

To reduce confusion, **the word "lime" is replaced with "green"** in the revised manuscript.

The criterion of "close to zero" is a little confusing indeed. In short, we selected the bottom 5% absolute VTECP$'$ values in selected areas according to different plots. We had planned to give a detailed interpretation in section 3.2 with Figure 7 so we just gave a guidance of "see below for example". We have revised the text to make it clear.

According to this comment, the sentence

"The lime lines are similar but for pixels with values close to zero (see below for example)"

**is revised as**

**Page 6, Lines 10-11:** "The green lines are similar but for pixels with the bottom 5% absolute VTECP$'$ values in selected areas (see section 3.2 for a detailed example)."

**Comment 11**

Page 6, Lines 10-17, I suggest that the author have a look at the paper by Pradipta et al., (2016): Interhemispheric propagation and interactions of auroral traveling ionospheric disturbances near the equator. Journal of Geophysical Research: Space Physics, 121, 2462-2474, https://doi.org/10.1002/2015JA022043

**Reply 11**

Thanks for recommending this paper. We revised the statements and added some contents about the interhemispheric propagation of traveling ionospheric disturbances near the equator. **More detail discussions can be seen in Reply to Comment 17.**

**Statements below are added.**

**Page 6, lines 32-35:** "Besides, Pradipta et al. [2016] studied the interaction of the auroral LSTIDs from opposite hemispheres near the dip equator during the 26 September 2011 geomagnetic storm. It shows that such interaction may bring much complexity to the TEC perturbations near the dip equator."

**Added Reference**

**Page 15, lines 29-31:** "Pradipta, R., Valladares, C. E., Carter, B. A., and Doherty, P. H.: Interhemispheric propagation and interactions of auroral traveling ionospheric disturbances near the equator. Journal of Geophysical Research: Space Physics, 121(3), 2462-2474, 2016."

**Section 1. Reply to the Review Comments #1**

**Comment 12**

Page 6, Lines 18-27, considering Figure 5 where there is a temporal shift seen at 38 degrees North followed by 29 degrees North, have the authors considered investigating such LSTID to be originating from northern hemisphere and propagation towards the equator with possibility of crossing the equator towards the southern hemisphere?

**Reply 12**

Thank you for this suggestion.

Yes, we did have considered it. We have plotted TLPs like those in Figure 8 for different longitudinal bands with spatiotemporal ranges of 07:00-14:00 UT and 15°S-45°N. It can be seen that there seems to be propagating perturbations that cross the dip equator.

**However, we decided not to include these rough results in our manuscript for two main reasons.** On the one hand, our current work mainly focuses on the northern hemisphere mid latitude. The perturbations in the low latitude are related to various mechanisms and thus much complex. We may investigate perturbations in this region in our future works. On the other hand, the collected TEC data has a relatively low spatiotemporal coverage in the low latitude and southern hemisphere of this longitudinal sector during this storm period.

**Comment 13**

Page 6, lines 28-44, In my view, ionosonde data and its interpretation should be given more attention than is done in the current version. If the uplift of virtual height (h') is due to the AGW which results into the TID that reaches ionospheric heights, then we would have seen the dominant trend in downward phase velocity. An important consideration with ionosonde data is to check that scaling was done correctly as I have already mentioned. In the current manuscript, the authors paid too much attention to the higher iso-line and connected it to the one at the lower h'. In my opinion, this should be re-looked at, because the ionosonde is 'more accurate' at measuring the bottom-side ionosphere. Although I don't know the actual height corresponding to virtual height of 600 km, it may be possible that this could fall within the extrapolated topside? Therefore, in actual sense, we should be able to see the downward phase velocities from the lower h' values for the analysis to be reliable. For a recent analysis of ionosonde data during this storm period, please see https://doi.org/10.1002/2017JA024510.

**Reply 13**

Thanks for the suggestions about the ionosonde data.

This comment is related to the Comment 4 about the error checking. The virtual heights of the iso-frequencies were scaled from the consecutive ionograms with 15-minute sample rate. All the virtual heights are below the virtual height of the h'F2. In order to obtain more useful data, the virtual heights near the peak F2 layer were also scaled and given in the original manuscript. The data scaling is done by ourselves with the professional scaling software provided by CRIRP. During the revision, we have checked this process again and thus we tend to believe that the data is credible. But as the reviewer point out, the trend in downward phase velocity is sometime not clear. We think this may be attributed to the low temporal resolution of 15 minutes.

Another issue that the reviewer put forward is that we might pay too much attention to the higher iso-line and the ionosonde is "more accurate" at measuring the bottom-side ionosphere. We check the iso-frequency lines and find that just as the reviewer point out, the downward phase velocity, if it can be detected in a ionosonde station, mainly occurs in lower h' (frequency). So considering this fact, we revise the iso-frequency plots by limiting the frequency to be less than 7

MHz. The text is also revised accordingly.

**The revised content about this paragraph and the revised figure is as follows:**

Page 7, lines 6-24: "Ionospheric parameters from ionograms have been commonly used since early TID studies. Recently, ionograms and iso-frequency lines with different sampling rates were used in TID studies [Klausner et al, 2009; Ding et al., 2012, 2013; Pradipta et al., 2015;

Ramsingh et al., 2015; Habarulema et al., 2018]. Figure 6 presents the temporal variations of the virtual height for each iso-frequency line. The names and locations of the corresponding ionosondes are given in each subplot as annotates. The sampling frequency are marked on the right side for each line. On the left column, the results of five stations are arranged in order from high to lower latitudes, and on the right column, it shows the recordings of four stations in the same latitudinal belt. …… Such trends (marked with black dashed lines) indicate a downward vertical phase velocity, which is one of the typical characteristics of TID and AGW [Hine, 1960;

Hocke and Schlegel, 1996]. It should be noted that the downward trend is not much clear for certain station, especially the one in Qingdao. This may be attributed to the 15 minutes sampling interval."

[Figure]

**Comment 14**

Page 7, lines 11-12, the text which talks about morphology changes of the TID changing as it propagates from high to lower latitudes: Is this backed by any references? Because the analysis of the authors is limited to northern hemisphere mid-latitudes

**Reply 14**

Thank you very much.

The sentence in the original manuscript can be misunderstood. Here with this expression we just want to mention the TID morphology changes with time that shown in Figure 4. In the revised manuscript, we try to make this clearly as follows: .

**Page 7, lines 35-36:** "It should be noted that the morphology of this TID is continuously changing as it moves from high to lower latitudes in the studied region."

**Comment 15**

Page 7, lines 15-20: The authors can consider labelling the 'rectangles' as A, B, C, etc or something along this line for the reader to easily identify them in the Figure.

**Reply 15**

Thanks for this suggestion. The labels are added and the text is revised accordingly as follows:

**Page 7, Lines 41-42 ~ Page 8, Lines 1-2:** "…, which are marked with dashed rectangles A-F in Figure 7. For each band, the VTECP′ data is averaged along the latitude for every 6 minutes (0.1 hours), and the results as a function of UT and latitude are illustrated correspondingly in Figure 8 (a-f)."

[Figure]

[Figure]

**Comment 16**

Page 7, lines 37-38: This is where a discussion/description of the agreement between the two techniques (VTECP' and doppler radar) should have discussed. Please refer to my earlier comment in the opening statements.

**Reply 16**

Thanks for this thought-provoking suggestion.

Based on this comment and the Comment 7, we have revised the estimated speed (535 m/s) from Doppler recordings. To show the good agreement between this speed and the VTECP speed, we made a estimation of the VTECP speed specifically for the reflecting points' longitudinal band (111°E-114°E, 29°N-38°N). Related discussions have been added as a separate paragraph.

**The original statement:**

"It shows that Vt is in good agreement with the result of 553 m/s derived from the Doppler observation. These parameters are typical for an LSTID. Besides, it is interesting to note that the mean Vc is slightly larger than the mean Vt, which seems like the wave behind is pushing that ahead."

**is revised as:**

**Page 8, Lines 28-43 ~ Page 9, Line 1:** "These parameters are typical for an LSTID. Vt and Vc overlap, although only marginally, considering the error ranges. Meanwhile, the mean Vc is slightly larger than the mean Vt, which seems like the wave behind is pushing that ahead. In general, the speed of trough and crest of the LSTID should be rather the same since they are induced by the same gravity wave. However, the wave properties might change with time dependent on the forcing from background condition, especially for LSTID covering large spatial region. This might explain the differences.

In addition, it is interesting to note that Vt is in reasonable agreement with the result of 535 m/s derived from the Doppler recordings. To show it more specifically, we estimated the speed and direction of the LSTID using the same TLP method as Figure 8 but in 111°E-114°E and 29°N-38°N (corresponding to the reflecting points). The result is 562±59 m/s and 0°, respectively. In general, the LSTID velocity estimated from ground-based stations tend to be larger than the actual velocity since these stations, in most cases, are not in perfect alignment with the propagation direction of the LSTID [Afraimovich et al., 1998; Habarulema et al., 2013]. Such good agreement between VTECP' and HF Doppler results may be attributed to the fact that the reflecting points (29.2°N,111.8°E; 38.0°N,113.2°E) of the Doppler receivers are in a narrow longitudinal band and the direction of the LSTID's propagation is also almost due south between 111°E-114°E."

**Comment 17**

Page 8, line 6, after Chimonas, 1970; add a reference https://doi.org/10.1002/2016GL069740 as these authors directly reported related results based on GNSS TEC observations and other measurements. Very recently, Jonah et al., (2018), available on https://doi.org/10.1029/2018JA025367, reported related results during storm conditions. Consult this reference as well (I think their Figure 4). On this point, https://doi.org/10.1002/2017JA024510

reported equatorward LSTID propagating from the southern hemisphere crossing to the northern hemisphere in the Asian region during this storm period. In fact, their analysis showed that these TIDs may not have exceeded 30 degrees North, which may be in agreement with your analyses and is more clearer in Figure 8(c) at around 1200 UT. Please have a look at their Figure 3(e) and possibly add some discussion to this effect.

**Reply 17**

Thank you for the recommended references. They are added in the text and the logical structure of the text is changed accordingly.

**The sentence in the original manuscript**

"Besides, the perturbations at 20°N around 12:00 UT and 13:00 UT show patterns of poleward movement. Ding et al. [2013] have studied the poleward-propagating LSTIDs in southern China during a medium-scale storm in 2011. They attribute their observations to the excitation of secondary LSTIDs during the dissipation of primary disturbances from the lower atmosphere. In addition, the poleward-moving disturbances may also be induced by the variation of the equatorial electrojet [Chimonas, 1970] or just propagate from the southern hemisphere [Zakharenkova et al., 2016]. A detailed investigation of this phenomenon is not the focus of this work."

**is revised as:**

**Page 9, Lines 6-18:** "Besides, the perturbations at 20°N around 12:00 UT and 13:00 UT show patterns of poleward movement. Habarulema et al. [2018] have identified TIDs in the Asian-Australian sector during the same storm period. It provides clear examples of TIDs crossing the dip equator from the southern hemisphere to the northern hemisphere around 09:00-12:00 UT. Their analysis shows that these TIDs may not have exceeded 30°N. Such poleward feature is also detected in other longitudinal sectors during this storm [Zakharenkova et al., 2016] and other storms [Pradipta et al., 2016; Jonah et al., 2018]. In addition, Ding et al. [2013] have studied the poleward-propagating LSTIDs in southern China during a medium-scale storm in 2011. They attribute their observations to the excitation of secondary LSTIDs during the dissipation of primary disturbances from the lower atmosphere. Besides, the poleward-moving disturbances may also be induced by the variation of the equatorial electrojet as pointed out by Chimonas [1970] and more recently by Habarulema et al. [2016]. A detailed investigation of this phenomenon is not the focus of this work."

**New added References**

**Page 14, Lines 39-42:** Jonah, O. F., Coster, A., Zhang, S., Goncharenko, L., Erickson, P. J., Paula, E. R., and Kherani, E. A.: TID observations and source analysis during the 2017 Memorial Day weekend geomagnetic storm over North America. Journal of Geophysical Research: Space Physics, 123, 8749– 8765. https://doi.org/10.1029/2018JA025367, 2018.

**Accordingly, some statements in the Discussion section are rephrased since new references are included. For example:**

"Zakharenkova et al. [2016] have studied the behaviors of LSTIDs during the St. Patrick's Day storm for the European and American sectors with GPS and GLONASS observations. It shows clearly in their results that the European sector (10°E) also exhibits LSTIDs around 11:00 UT. … The fitting lines are obtained with a similar method mentioned above.

Figure 10 is basically consistent with the Figure 3(b) in Zakharenkova et al. [2016], such as …”

**is revised as:**

**Page 10, Lines 38-41 ~ Page 11, Lines 5-7:** “During the 2015 St. Patrick's Day storm, LSTIDs in the European-African, American and Asian-Australian sectors are detected and analysed with TEC observations [Borries et al., 2016; Zakharenkova et al., 2016; Habarulema et al., 2018]. It shows clearly in their results that the European sector also exhibits LSTIDs around 11:00 UT. …… The fitting lines are obtained with the same method as those in Figure 8.

Figure 10 is basically consistent with previous results, such as ……”

**Summary:**

**Comment 18**

Page 10, lines 1-3: The authors may want to rephrase this statement given that an earlier study by Habarulema et al., (2018)– https://doi.org/10.1002/2017JA024510 - provided some analysis for this particular storm in the Asian sector. May be the analyses was not as detailed as provided in this paper, but definitely this is not the first analysis for this storm in the Asian region. The strength of this paper over what was presented in Habarulema et al., (2018) and other attempted studies is the use of multiplicity of data sources to provide more details and clarity during this storm period.

**Reply 18**

Thanks for pointing out this inappropriate statement. We have revised our text accordingly.

“Using data from 4 GPS receiver networks (CMGN, CMONOC, GEONET, IGS), together with recordings of 2 HF Doppler shift stations and 8 ionosondes, we show the first observation results of the LSTIDs in the East Asian sector during the 2015 St. Patrick's Day storm.”

**is revised as:**

**Page 11, Lines 25-28:** “Using data from 4 GPS receiver networks (CMGN, CMONOC, GEONET, IGS), together with recordings of 2 HF Doppler shift stations and 8 ionosondes, we provide comprehensive and detailed observation results of the LSTIDs in the East Asian sector during the 2015 St. Patrick's Day storm.”

**Reply to the Review Comments #2**

**The manuscript addresses the interesting scientific problem of understanding the properties of large scale travelling ionospheric disturbances (LSTIDs), which are frequently observed during geomagnetic storms. This manuscript discusses the properties of LSTIDs during 17th March 2015 with focus on the Chinese and Japanese sector. Although, descriptions of LSTID occurrence during this event have been published before, this manuscript adds new aspects on the longitudinal dependence of the LSTID properties in the Chinese/ Japanese sector based on GNSS, HF and ionosonde data. The manuscript is well structured, well written and presents analysis of high quality in a well understandable way. Thus, my overall evaluation is publishing after solving minor remarks. The manuscript in its current form has three weak points.**

Thank you for your substantial and detailed comments here and in the supplement material! According to these comments, we revised the manuscripts and gave our replies to these comments point by point. The original manuscript, the revised version and the added references are listed if necessary. The red colors mark the revised parts and the new references.

Note that this reply is focusing on the Review Comments #2. The revised parts and the new references according to the Review Comments #1 are marked with yellow highlights when there are overlaps.

**Reply to Major Comment:**

**Major Comment I.**

**I. First, already in the abstract the authors are referring to negative and positive LSTIDs and seem to treat them in the course of the manuscript like separate phenomenon. Since these LSTIDs are the signature of atmospheric gravity waves, both signatures belong to the same wave. Therefore, I would recommend to avoid discussing positive and negative amplitudes separately.**

**Related Annotation****: measuring wave trough and crest should be rather the same, since they belong to one phenomenon, which is the gravity wave. However, the wave properties might change with time dependent on the forcing. This might explain the differences of both measurements. Looking at the error margins, both Vt and Vc indeed overlap (although only marginally). This is good.**

**Reply I.**

Thank you very much for these suggestions!

Firstly, we misused phrases of "negative and positive LSTIDs". They should be "the trough and crest of the LSTID". This has also been pointed out by the Review Comments #1. We have revised those misleading statements accordingly. Besides, because the scale of this LSTID (period and wavelength) is very large and cover large spatial region, the wavefront of the LSTID maybe deformed due to different background condition during its propagation from higher to lower latitude, we discussed the trough and crest of the LSTID separately since they behave differently, which can be seen in our results. This may be attributed to, as the reviewer suggest, that the wave properties change with time dependent on the forcing. Meanwhile, Vt and Vc show certain consistency. We have revised our manuscript accordingly to make these clear. To be specific:

**The statement**

"Besides, it is interesting to note that the mean Vc is slightly larger than the mean Vt, which seems like the wave behind is pushing that ahead."

**is revised to**

**Page 8, Lines 28-33:** "Vt and Vc overlap, although only marginally, considering the error ranges. Meanwhile, the mean Vc is slightly larger than the mean Vt, which seems like the wave behind is pushing that ahead. In general, the speed of trough and crest of the LSTID should be rather the same since they are induced by the same gravity wave. However, the wave properties might change with time dependent on the forcing from background condition, especially for LSTID covering large spatial region. This might explain the differences."

**Major Comment II.**

**II. Second, the key point of the manuscript is the discussion of longitudinal dependence of LSTID properties. But, this is impacted by the data coverage. The data coverage is lower in the east and west boundaries of the investigated region. I argue that this impacts the accuracy of the estimation of the LSTID properties. The discussion of the LSTID properties (wavelength, period and speed) should be treated with more care concerning reliability of the results.**

**Related Annotation.** **Please elaborate on the impact of data coverage and size of the regions of investigation on the accuracy of the results. The regions used for the Time-Latitude Plots cover about 20° in latitude. This is roughly one wavelength. If the data coverage is reduced (what is the case in the East and West regions), this introduces certainly an error on the results.**

**Related Annotation.** **Nice figure and good illustration. Again, I am requesting to elaborate on the accuracy of the results. The deviation between geomagnetic declination and wave propagation direction is largest in the East and West, where you have lower data coverage.**

**Related Annotation.** **I see it critical to highlight this fact, on the one hand because there is no clear dependence and explanation for the longitudinal tendency and on the other hand, it might be impacted by the measurement properties.**

**Reply II.**

Thank you very much!

Indeed, the data coverage in the East and West boundaries of the studied region is relatively lower comparing to that between 100°E-120°E. Such difference in data coverage is resulted from both GPS-receivers and land-sea distributions. Besides, the studied latitudinal range is ~ 20°, which is roughly one wavelength as the reviewer pointed out.

Meanwhile, it should be noted that we do had considered these issues in the manuscript and tried our best to reduce such influences. For example, we selected areas for every l0 longitudinal degrees with varying latitudinal ranges (Figure 7) to include as many data points as possible. Data in longitudinal bands of 70°E-80°E and 140°E-150°E were not used considering the bad data coverage. Besides, data in every 0.1-hours bin in TLPs (Figure 8) was obtained by latitudinal average for every 10° longitudinal band. We find that data in 130°E-140°E is hugely influenced by data coverage, so it has not been used when deriving LSTID parameters. In addition, since the studied latitudinal range is roughly one wavelength, it is hard to estimate the wavelength directly from the 2D VTECP map. Instead, the wavelength is estimate with the velocity and period from TLPs.

Of course, these cannot totally eliminate the data-coverage influence to the estimation accuracy of the LSTID parameters. Our results should be examined by further studies with better data coverage in a wider longitudinal range.

**The statement**

"Finally, with the period and speed, the wavelength can be easily determined."

**is revised to**

**Page 8, Lines 17-20:** "As for the estimation of wavelength, note that the studied area is ~ 20° in latitude, which is roughly one wavelength and thus make it difficult to estimate the wavelength directly from the 2D VTECP′ map. So, the wavelength is derived from the multiplication of speed and period."

**The statement**

"The longitudinal dependence of these parameters can be seen clearly."

**is revised to**

**Page 8, Lines 23-26:** "It can be seen that these parameters show certain longitudinal dependence. It should be noted that the data coverage is relatively lower in the east and west boundaries of the investigated region. This may impact the accuracy of the estimation of the LSTID properties in these areas."

**The statement**

"It should be noted that our speculation needs to be verified with more observational data and numerical simulation to reduce uncertainty in our propagation estimation and to figure out the detailed physical processes."

**is revised to**

**Page 10, Lines 34-37:** "Besides, considering the relatively low data coverage in the East/West side of the studied region, it should be noted that our speculation needs to be verified with more observational data and numerical simulation to reduce uncertainty in our propagation estimation and to figure out the detailed physical processes."

**The statement**

"(3) Other propagation parameters are also longitudinal dependent (see Table 1), and the mean values and standard deviations of the period, Vt, Vc, and wavelength are 74.8±1.4 minutes, 578±16 m/s, 617±23 m/s, and 2691±80 km, respectively.".

**is revised to**

**Page 11, Lines 40-41 ~ Page 12, Lines 1-6:** "(3) The propagation parameters in different longitudinal bands are estimated. These parameters show certain longitudinal dependence. Besides, the mean values and standard deviations of the period, Vt, Vc, and wavelength are 74.8±1.4

minutes, 578±16 m/s, 617±23 m/s, and 2691±80 km, respectively.

It should be noted that our results show certain consistency with previous works focusing on the

Chinese or Japanese sector for different LSTID events. Nevertheless, the longitudinal dependence shown in our results should be examined further with more case studies based on large longitudinal and high-resolution coverage of GPS data.”

”.

**Major Comment III.**

**III. Third, the authors present also the result of the LSTID occurrence in Europe. This has**

**been extensively discussed in Borries et al. (2016, https://doi.org/10.1002/2016JA023178).**

**Specifically, the LSTID occurring between 11 and 12 UT has been discussed to be**

**“special” because it is impacted by winds and prompt penetration electric fields at the**

**same time. This fact should be included in the discussion of this manuscript. It supports**

**the finding of the authors that the LSTID properties in Europe differ from the LSTID**

**properties in the Chinese/ Japanese sector.**

**Related Annotation: Borries et al. (2016) shows and discusses some more details about the**

**LSTIDs. The LSTID between 11 and 12 UT is specifically large and a combination of**

**wind and electric field effects.**

**Related Annotation: The reason for the difference are the different physical processes that**

**compete over Europe (as described in Borries et al., 2016).**

**Related Annotation: It is not clear why these LSTIDs can be only driven by winds. Please**

**elaborate what would be the difference in case these TEC perturbations would be**

**driven by electric fields.**

**Reply III.**

Thank you very much for recommending this reference!

Actually, after the manuscript submission, we noticed that we missed this reference. This paper has also been recommended by the Review Comments #1. We have already added it into the revision.

According to this comment and related annotation, the statements about the difference between East-Asian and European sectors focusing on PPEF are added accordingly.

**The statement**

“In addition, the tendency of field-aligned propagation of the LSTID indicates that it is driven by the neutral winds rather than by electric fields since the winds push the plasma up and down along the magnetic field lines.”

**is revised to**

**Page 10, Lines 29-34:** “In addition, the tendency of field-aligned propagation of the LSTID

indicates that it is driven by the neutral winds since the winds push the plasma up and down along the magnetic field lines. There is no evidence, such as simultaneous perturbations at all latitudes in other cases [Borries et al., 2016; Zakharenkova et al., 2016], to show that the LSTID in the

Chinese/Japanese sector is affected by prompt penetration electric field (PPEF) during the same period."

**Statements below are added.**

**Page 11, Lines 15-19:** "Borries et al. [2016] present a detailed study on the LSTID in Europe during this storm. It is suggested that the perturbation occurring around 11:00 UT is special since it is impacted by PPEF and wind at the same time. Comparatively, the LSTID in the

Chinese/Japanese sector seems only driven by winds. This may partly account for the longitudinal difference in our results."

# Technical Recommendations/Corrections:

**Comment 1**

**Jakowski et al. (2008) did not discuss LSTIDs but large scale gradients (no discussion of**

**wave properties)**

**Reply 1**

Thank you very much! **This reference has been deleted** in the revised article.

**Comment 2**

**URLs of the data sources are usually provided in the acknowledgements, not in the text.**

**Reply 2**

Thank you very much! URLs of the data sources have been **moved into the acknowledgements**.

**Comment 3**

**Presenting the LSTID results with VTECP' has the advantage of better illustrating most**

**wave properties, but it does not represent the true wave amplitude anymore. On a quick**

**view, the figures might be misinterpreted. Therefore, I recommend to make it very clear**

**that this is an "artificial" amplitude.**

**Related Annotation: This is certainly nice to better visualize the positive and negative**

**amplitude of the TEC perturbation. But the authors should take care, that the readers**

**do not misinterprete the amplitude of the TEC perturbation. It looks like the Amplitude**

**is about 2 TECU, but is that the true amplitude?**

**Reply 3**

Thank you very much for this suggestion! We have revised the manuscript accordingly.

**The statement**

"Note that the raw value of VTECP is converted into VTECP' with

$$VTECP' = sign(VTECP) * log_{10}(abs(VTECP) + 1)$$

to make it easier to distinguish the regions with positive and negative perturbations."

**has been revised to:**

**Page 6, Lines 2-8:** "The raw value of VTECP has already been converted into VTECP' with the equation

$$VTECP' = sgn(VTECP) * log_{10}(abs(VTECP) + 1) \qquad (3)$$

The raw amplitude of VTECP above 30°N is ~ 2 TECu while the raw amplitude of VTECP below 30°N reaches ~ 10 TECu. So, transform (3) provides a better colormap for 2D VTECP plots by sharpening the edges between positive and negative values and reduce the differences of VTECP in middle and low latitudes. Consequently, it should be noted that the amplitude of the wavelike variation does not represent the true wave amplitude but an "artificial" one."

**The Figure 4 caption**

"**Figure 4.** A series of 2D VTECP′ maps over the East Asian sector from the period of 09:40-09:50 UT to 11:30-11:40 UT on 17 March 2015. The grey areas represent the nightside. The colorbar represents the VTECP′ (units: TECu). The lime and yellow lines illustrate the least square fittings (order 2) for wavefronts."

**is revised as**

**Page 17, Lines 13-17:** "**Figure 4.** A series of 2D VTECP′ maps over the East Asian sector from the period of 09:40-09:50 UT to 11:30-11:40 UT on 17 March 2015. The grey areas represent the nightside. The colorbar represents the VTECP′ (units: TECu), which is transformed from the original VTECP value with equation (3) for a more viewer-friendly colormap. The green and yellow lines illustrate the least square fittings (order 2) for wavefronts."

**Comment 4**

**Figures 3 and 5 do not have much content. But they are supposed to be compared with each other. Therefore, I recommend to join the content of both figures into one figure. This will increase the information density and allow better comparability.**

**Related Annotation: I think, it would be nice to join Fig3 and Fig 5 to one plot, to have a more direct comparison of peaks in both plots and better visualize the similarities.**

**Reply 4**

Thank you very much!

It is a good idea to combine Figure 3 and 5 together to have a more direct comparison. However, combining them together will change the logical frame of the manuscript and a lot of places need to be rewritten considering the context. So, we plot the variation of doppler shifts into Figure 5.

**Figure 5 is revised to**

[Figure]

**Figure 5 description is expanded to**

**Page 17, Lines 18-20:** "**Figure 5.** Temporal variations of mean VTECP′ near the Doppler reflection points between 08:00 UT and 14:00 UT, 17 March 2015. Doppler shift recordings in Figure 3 are plotted with dashed lines for comparison."

**The statement below is added.**

**Page 6, lines 38-39:** "Doppler shift recordings in Figure 3 are also plotted with dashed lines for comparison."

**Comment 5**

**In figure 8, the impact of EIA is addressed. I assume, the dashed black lines indicate the boundary of EIA. This should be made clear in the text and figure description**

**Related Annotation: Please include a sentence describing, how the reader is going to identify the EIA region. I assume, the black dashed lines are encapsulating this region.**

**Reply 5**

Thank you very much! The manuscript has been revised accordingly.

**The statements**

"As mentioned before, the variation of VTECP′ in the EIA region is rather complex, so only the values over 30°N (marked with dashed lines) are used to estimate the speed."

**is revised to**

**Page 8, Lines 2-5:** "As mentioned before, the VTECP′ variation related to EIA is rather complex. Considering that EIA is mainly a low-latitudinal phenomenon, the 30°N is marked with black dashed lines in Figure 8 which indicate the boundary of EIA. Only values over 30°N are used to estimate the speed."

**Figure 8 caption**

"**Figure 8.** TLPs of VTECP′ for different longitudinal bands between 07:00-14:00 UT. White dots give the data points for linear fitting, and the fitting results are marked with white lines. Black dashed lines depict 30°N in (b-d, f) and 40°N in (f)."

**is revised as**

**Page 17, Lines 28-31:** "**Figure 8.** TLPs of VTECP′ for different longitudinal bands between 07:00-14:00 UT. White dots give the data points for linear fitting, and the fitting results are marked with white lines. 30°N in (b-d, f) is marked with black dashed lines which indicate the boundary of EIA. 40°N is marked in (f)."

**Comment 6**

**In the discussion section, the authors exclude the impact of electric fields on the LSTID propagation and favour the impact of winds, driving the LSTID propagation because of field-aligned propagation. For a better understanding, the authors should explain, what would be different in case of electric field impact. In fact, since Borries et al. (2016) describe prompt penetration electric field impact in Europe at that time, more emphasis should be given to discuss electric field impact in the Chinese/Japanese sector at the same time.**

**Reply 6**

Thank you very much for this suggestion! This Comment is related to the Major Comment III. Please refer to replies to in that section in page 2.

**Section 2. Reply to the Review Comments #2**

**Comment 7**

**I detected a few spelling errors and grammar issues (indicated in the supplementary material). I expect, there are more than I found and recommend professional editing.**

**Reply 7**

Thank you very much!

We have revised all the spelling errors and grammar issues that indicated in the supplementary material. We have further checked throughout the article and found several other errors and they have also been revised in the revision. All of them are marked with red color in the supplement material.

**Section 3. Marked-up Manuscript**

[revised manuscript text omitted]

---

## Referee Report (RR1)

Comments on revised manuscript "**A case study of the large-scale traveling ionospheric disturbances in the East Asian sector during the 2015 St. Patrick's Day geomagnetic storm**" by Liu et al 2019

The authors have addressed all the comments comprehensively and the manuscript is significantly improved. There are still some missing words and I would like to suggest that careful language proofreading and editing is done by the authors.
The manuscript is acceptable and from my reading, I don't think that it needs another round of review.
I have one minor comment that I leave for the authors to decide whether to consider or not.

Lines 15-17: The text which talks about trough and crest. I suggest that the authors modify the statement to "Results show a LSTID spanning at-least 60 degrees in longitude (60ºE-140ºE) occurs as a result of possibly AGWs propagating from high to lower latitudes around 09:40 UT-?? … (here the end time should also cover the crest of the LSTID)"

Line 18, "...high frequency Doppler shift ...", shouldn't the word "shift" be "radar"?

---

## Author Response (AR2)

**Reply to Review Comments**

**Dear Editor,**

Thank you very much for reviewing this manuscript, and thank the anonymous reviewer for the second round of review.

We did our best to revise the manuscript according to the review comments. We proofread our manuscript carefully, and corrected some errors of the word and grammar. We also revised some confusing statements.

Our replies to the 2 detailed comments are as follows.

**Comment 1**

**Lines 15-17: The text which talks about trough and crest. I suggest that the authors modify the statement to "Results show a LSTID spanning at-least 60 degrees in longitude (60° E-140°E) occurs as a result of possibly AGWs propagating from high to lower latitudes around 09:40 UT-?? ⋯ (here the end time should also cover the crest of the LSTID)"**

**Reply 1**

Thank you very much!

According to this comment, **the statement in the manuscript**

"Results show that a trough of LSTID spanning at least 60° in longitude (80°E-140°E) occurs and propagates from high to lower latitudes around 09:40-11:20 UT. It is followed by a crest of LSTID which shows a tendency of dissipation starting from the East side."

**is revised to:**

**Page 3, Lines 14-17:** "Results show an LSTID spanning at least 60 degrees in longitude (80°E-140°E) occurs as a result of possibly AGWs propagating from high to lower latitudes around 09:40-11:40 UT, and the crest of this LSTID shows a tendency of dissipation starting from the East side."

**Correspondingly, similar statements in the Summary**

"A trough of LSTID occurs and propagates from high to lower latitudes during 09:40-11:20 UT, which spans over 60° in longitude. It is followed by a crest of LSTID that characterized by a clear tendency to dissipate starting from the East side."

**is revised to:**

**Page 13, Lines 28-30:** "An LSTID occurs as a result of possibly AGWs propagating from high to lower latitudes around 09:40-11:40 UT, which spans over 60° in longitude, and the crest of this LSTID is characterized by a clear tendency to dissipate starting from the East side."

**Comment 2**

**Line 18, "...high frequency Doppler shift ...", shouldn't the word "shift" be "radar"?**

**Reply 2**

Thank you very much!

In order to avoid readers' confusion about our devices and HF Radar, we did not use the term "radar" in the last revised manuscript. We referred some papers concerning the HF Doppler shift record, and find the "HF Doppler sounder" is more suitable here, so:

**The statement in the manuscript**

"… the recordings from 2 high-frequency Doppler shift stations and …"

**is revised to**

[revised manuscript text omitted]